# Collagen prolyl 4-hydroxylase 1 is essential for HIF-1α stabilization and TNBC chemoresistance

Gaofeng Xiong[1,2], Rachel L. Stewart [3], Jie Chen[1,2], Tianyan Gao[1,4], Timothy L. Scott[5,6], Luis M. Samayoa[3], Kathleen O'Connor[1,4], Andrew N. Lane[5,6] & Ren Xu [1,2]

Collagen prolyl 4-hydroxylase (P4H) expression and collagen hydroxylation in cancer cells are necessary for breast cancer progression. Here, we show that P4H alpha 1 subunit (P4HA1) protein expression is induced in triple-negative breast cancer (TNBC) and HER2 positive breast cancer. By modulating alpha ketoglutarate (α-KG) and succinate levels P4HA1 expression reduces proline hydroxylation on hypoxia-inducible factor (HIF) 1α, enhancing its stability in cancer cells. Activation of the P4HA/HIF-1 axis enhances cancer cell stemness, accompanied by decreased oxidative phosphorylation and reactive oxygen species (ROS) levels. Inhibition of P4HA1 sensitizes TNBC to the chemotherapeutic agent docetaxel and doxorubicin in xenografts and patient-derived models. We also show that increased P4HA1 expression correlates with short relapse-free survival in TNBC patients who received chemotherapy. These results suggest that P4HA1 promotes chemoresistance by modulating HIF-1-dependent cancer cell stemness. Targeting collagen P4H is a promising strategy to inhibit tumor progression and sensitize TNBC to chemotherapeutic agents.

[1] UK Markey Cancer Center, University of Kentucky, Lexington, KY 40536, USA. [2] Department of Pharmacology and Nutritional Sciences, University of Kentucky, Lexington, KY 40536, USA. [3] Department of Pathology and Laboratory Medicine, University of Kentucky, Lexington, KY 40536, USA. [4] Department of Molecular and Cellular Biochemistry, University of Kentucky, Lexington, KY 40536, USA. [5] Center for Environmental and Systems Biochemistry, University of Kentucky, Lexington, KY 40536, USA. [6] Department of Toxicology and Cancer Biology, University of Kentucky, Lexington, KY 40536, USA. Correspondence and requests for materials should be addressed to R.X. (email: ren.xu2010@uky.edu)

Prolyl hydroxylation, a common post-translational modification, modulates protein folding and stability in mammalian cells. The abundance of hydroxyproline among the residues in animal proteins is about 4%, and most of the hydroxyproline is found within the collagen[1,2]. Collagen prolyl 4-hydroxylase (P4H) is an $\alpha_2\beta_2$ tetrameric $\alpha$-ketoglutarate ($\alpha$-KG)-dependent dioxygenase that catalyzes 4-hydroxylation of proline to promote formation of the collagen triple helix, releasing succinate as a product[3]. The P4H $\alpha$subunit (P4HA) is responsible for both peptide binding and catalytic activity. This process can be blocked by a number of inhibitors. Three P4HA isoforms (P4HA1-3) have been identified in mammalian cells[2]. P4HA1 is the major isoform in most cell types and tissues, and contributes to the majority of the prolyl 4-hydroxylase activity[4]. Increased collagen production is associated with breast cancer development and progression, and stromal cells are the major source of collagen deposition[5,6]. The expression of collagen P4H is significantly upregulated during breast cancer development and progression, and increased P4HA expression correlates with poor prognosis[7,8]. Interestingly, induction of P4HA1 expression in cancer cells is required for breast cancer metastasis[7]. However, we know little about how cancer cell P4HA1 promotes tumor progression.

High levels of hypoxia-inducible factor-1$\alpha$ (HIF-1$\alpha$) are associated with advanced cancer progression and poor clinical outcomes in breast cancer patients[9,10]. Activation of the HIF-1 pathway induces metabolic reprogramming and enhances angiogenesis, which is crucial for cancer progression[11,12]. De novo synthesized HIF-1$\alpha$ is rapidly hydroxylated by a family of oxygen-dependent dioxygenases (PHD) on proline 402 (Pro402) and proline 564 (Pro564)[13–15]. Proline hydroxylation induces HIF-1$\alpha$ ubiquitination and degradation, and subsequently reduces the half-life of HIF-1 protein[14,16]. The prolyl hydroxylation on HIF-1$\alpha$ is regulated by the concentration of the substrate oxygen[17,18]. Hyperactive HIF-1 pathway has been detected in triple-negative breast cancers (TNBCs)[19,20]. The differential activation of the HIF-1 pathway in breast cancer subtypes suggests that oxygen-independent pathways are involved in HIF-1$\alpha$ regulation during TNBC progression. However, the molecular mechanism underlying the HIF-1 activation in TNBC is not completely understood.

TNBC is an aggressive histological subtype with poor prognosis and accounts for approximately 15% of all breast cancer cases[21]. Patients with this cancer subtype have frequent metastases and a high rate of relapse after the first-line treatment[21–23]. Because TNBC is estrogen receptor (ER) negative, progesterone receptor (PR) negative, and Her2 negative, it is not responsive to hormone therapy and to drugs that target the HER2 protein. Chemotherapy regimens are standard of care treatment for TNBC, but more than 50% of patients are likely to experience cancer recurrence in the first 3 to 5 years after treatment[24]. Recent studies suggest that the activation of the HIF-1 pathway promotes chemoresistance in breast cancer[25,26]. Therefore, targeting the HIF-1 pathway is a potential strategy to suppress TNBC progression and chemoresistance.

Increased collagen deposition is associated with breast cancer development and progression, and stromal cells are considered the major source of collagen deposition[5]. Surprisingly, we and others have shown that increased expression of collagen prolyl 4-hydroxylase in breast cancer cells is required for cancer progression[7,8]. However, the critical molecular mechanisms that P4HA expression in cancer cells induces cancer progression have not been characterized. In the present study, we have identified a link between collagen hydroxylation and HIF-1 activation during TNBC progression. Our results suggest that inhibition of P4HA1 is a potential strategy to sensitize TNBC to chemotherapeutic agents.

## Results

**P4HA1 expression is associated with HIF-1 activation.** To define the roles of P4HA1 in breast cancer progression, we analyzed P4HA1 protein levels in human breast cancer tissue using tissue microarrays generated at UKY. We showed that P4HA1 expression was upregulated in TNBC and HER2-positive breast cancer tissues compared to the ER-positive breast cancer (Fig. 1a, b). P4HA1-positive staining was significantly enriched in high-stage TNBC tissues (Fig. 1c), indicating that P4HA1 expression is associated with TNBC progression. We also found that P4HA1 protein levels were increased in TNBC cell lines compared to luminal cancer cells (Fig. 1d). The upregulation of P4HA1 in breast cancer cell lines is associated with increased secretion of collagen (Fig. 1d).

Reducing P4HA1 expression or blocking its activity with P4H inhibitors (P4Hi) significantly represses tumor progression in the three-dimensional (3D) culture system (Supplementary Fig. 1a–f) and xenograft models[7,8]. To elucidate the molecular mechanisms by which P4HA1 promotes cancer progression, we performed unbiased gene co-expression and gene ontology analysis[27] using the breast cancer The Cancer Genome Atlas gene expression dataset. This analysis revealed that P4HA1 levels were positively correlated with gene expression signatures involved in response to hypoxia and glycolysis (Fig. 2a) (Supplementary Table 1). These data implicate that P4HA1 expression contributes to the hypoxic response and glycolysis during cancer development. HIF-1 is a key mediator of the hypoxic response and a pivotal regulator of the balance between lactic fermentation and pyruvate oxidation. We therefore tested whether the expression of P4HA1 induces the HIF-1 pathway in mammary epithelial cells. Introduction of exogenous P4HA1 with its cofactor P4HB increased HIF-1$\alpha$ protein levels in MCF10A and HMLE cell lines under both normoxia and hypoxia conditions (Fig. 2b, c). Quantitative reverse transcription-PCR (RT-PCR) data showed that P4HA1 expression had little effect on HIF-1$\alpha$ messenger RNA (mRNA) levels (Supplementary Fig. 2a), suggesting that the regulation is at the protein level. Knockdown of P4HA1 in TNBC cell lines reduced HIF-1$\alpha$ protein levels under both normoxia and hypoxia conditions (Fig. 2d, e). The expression of P4HA1 in P4HA1-silenced cells restored the HIF-1$\alpha$ levels (Supplementary Fig. 2b). We also found that secretion of collagen is regulated by P4HA1 in mammary epithelial cells (Supplementary Fig. 2c). In addition, treatment with P4HA inhibitor 1,4-dihydrophenonthrolin-4-one-3-carboxylic acid (1,4-DPCA) or ethyl-3,4-dihydroxybenzoic acid decreased HIF-1$\alpha$ protein levels in breast cancer cells (Fig. 2f).

To determine whether HIF-1 activity is regulated by P4HA1, we introduced the hypoxia response element (HRE)-luciferase reporter plasmid in control and P4HA1-silenced cells. The reporter data showed that knockdown of P4HA1 significantly reduced transcription driven by HRE (Fig. 2g), and treatment with the P4Hi also reduced the HRE-driven luciferase activity (Supplementary Fig. 2d). Next, we assessed expression of HIF-1 target genes lactate dehydrogenase A (LDHA) and pyruvate dehydrogenase kinase 1 (PDK1) in control, P4HA1-silenced, and P4HA1-expressing mammary epithelial cells. Quantitative RT-PCR data showed that silencing P4HA1 in breast cancer cells significantly reduced mRNA levels of LDHA and PDK1 under normoxia and hypoxia conditions (Fig. 2h). In contrast, expression of P4HA1 in MCF10A cells enhanced LDHA and PDK1 expression (Fig. 2i). In addition, P4HA1 expression significantly correlated with mRNA levels of LDHA and PDK1 in human breast cancer tissue (Supplementary Fig. 2e and 2f). These results indicate that P4HA1 expression in breast cancer cells induces activation of the HIF-1 pathway.

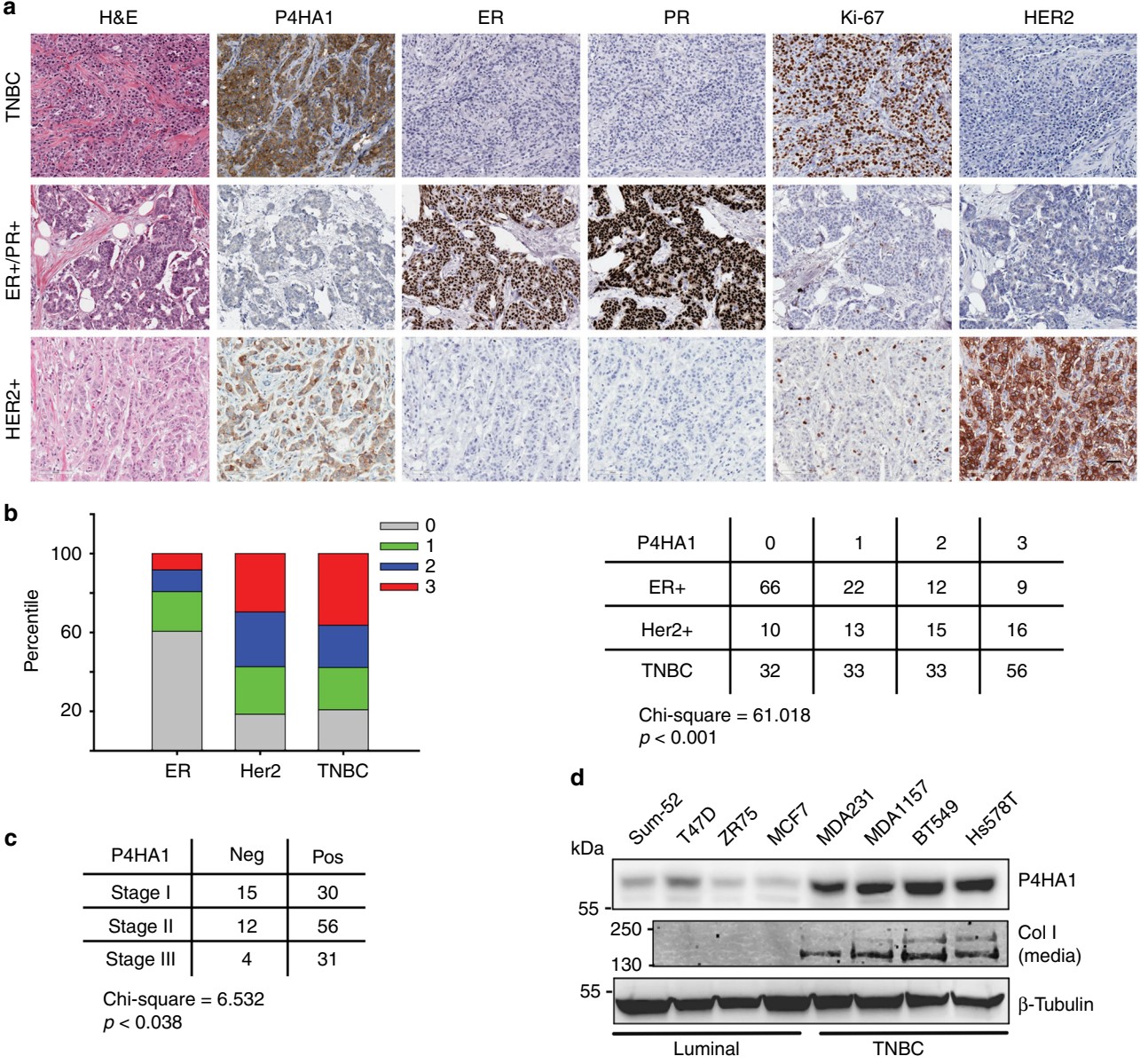

**Fig. 1** P4HA1 protein levels are significantly increased in TNBC. **a** Human breast cancer tissue array was stained by H&E and immunohistochemistry for ER, PR, P4HA1, Ki-67, and HER2. Bar: 40 μm. **b**, **c** Quantification of P4HA1 staining from **a**. The data were analyzed by $\chi^2$ test. **d** P4HA1 protein levels and collagen secretion were assessed in a panel of breast cancer cell lines

**Collagen hydroxylation enhances HIF-1α stability.** In the presence of active PHD, HIF-1α protein is hydroxylated and targeted for degradation by the ubiquitin–proteasome system[28], which dramatically reduces the half-life of HIF-1 protein[14,16]. Since P4HA1 expression had little effect on HIF-1α mRNA levels, we determined whether P4HA1 regulates HIF-1α protein stability. We performed a time-course experiment to determine HIF-1α protein degradation in control and P4HA1-silenced cells after cycloheximide treatment. We found that knockdown of P4HA1 significantly increased HIF-1α protein degradation at the early time points (Fig. 3a–d). HIF-1α ubiquitination was also upregulated in P4HA1-silenced cells (Fig. 3e). Importantly, treatment with the proteasome inhibitor Bortezomib rescued HIF-1α protein expression in P4HA1-silenced breast cancer cells (Fig. 3f). These results suggest that P4HA1 increases HIF-1α protein levels by reducing ubiquitination-mediated protein degradation.

Using the antibody against hydroxylated HIF-1α (P402), we showed that silencing P4HA1 drastically increased HIF-1α hydroxylation in MDA-MB-231 cells, while expression of P4HA1 and P4HB reduced proline hydroxylation in MCF10A cells (Fig. 3g). The ODD-luciferase reporter construct consisting of a firefly luciferase gene fused to the hydroxylation-dependent degradation region of HIF-1α has been used to monitor HIF-1α stability/degradation[29]. Using this reporter assay, we showed that expression of P4HA1 and P4HB significantly induced ODD-luciferase activity in MCF10A cells under normoxia and hypoxia conditions (Fig. 3h). In contrast, silencing P4HA1 reduced ODD-luciferase activity in MDA-MB-231 cells (Fig. 3i). It has been shown that point mutation on P402 and P564 on HIF-1α reduces proline hydroxylation and significantly increases protein stability; therefore, we introduced proline hydroxylation-deficient HIF-1α PPAA in P4HA1-silenced cells. We found that the HIF-1α PPAA mutant, but not wild-type HIF-1α, rescued HIF-1α expression

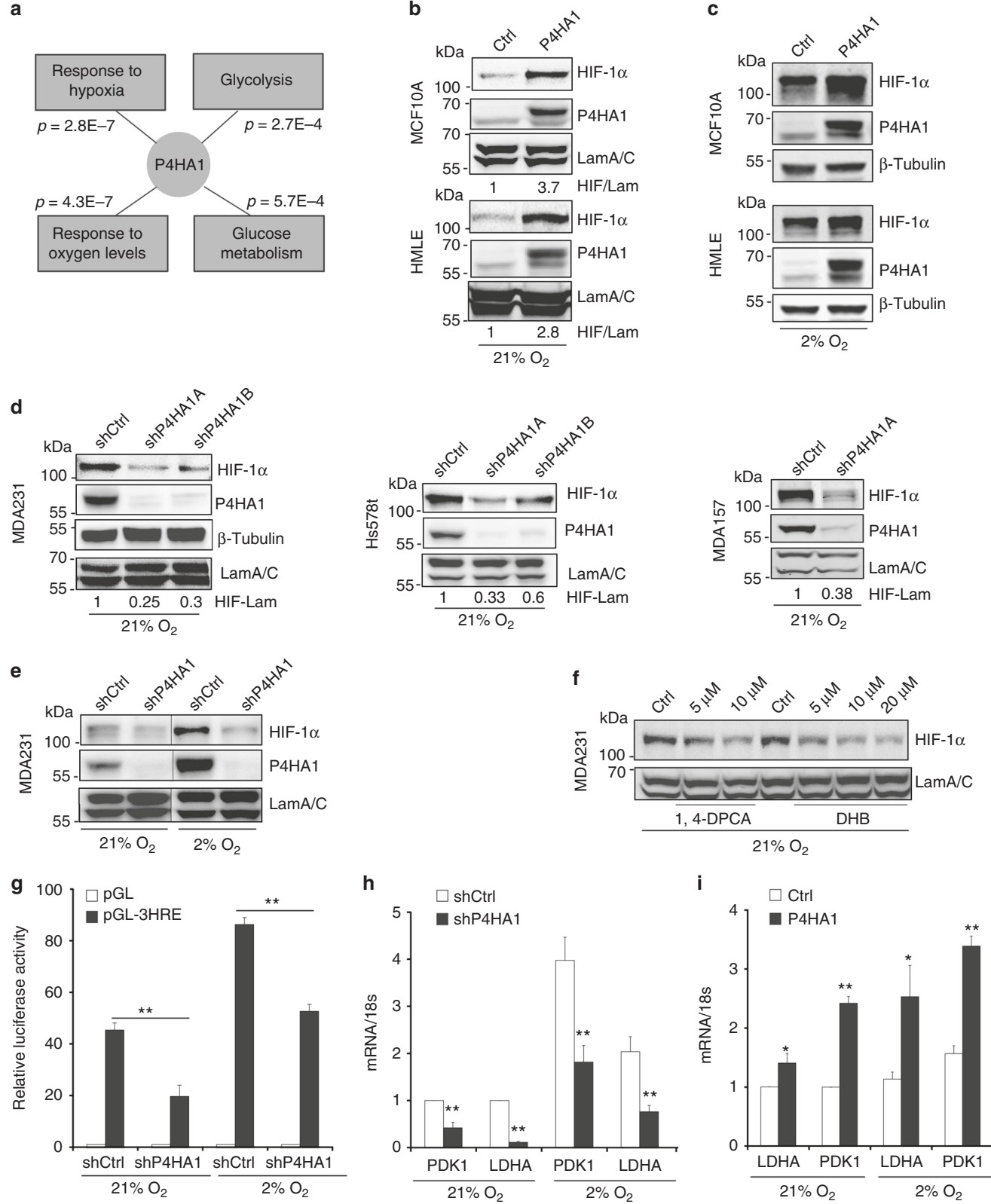

(Fig. 3j). These results strongly support the hypothesis that P4HA1 enhances HIF-1α protein stability by reducing proline hydroxylation.

We showed that silencing P4HA1 had little effect on the protein level of PHD1 and PHD3 and slightly reduced the PHD2 protein level (Supplementary Fig. 3a), suggesting that induced

HIF-1α hydroxylation is not due to increased PHD expression. In addition, silencing PHD2 but not PHD1 restored HIF-1α protein levels in P4HA1-silenced MDA-MB-231 cells (Supplementary Fig. 3b, c). These experiments indicate that P4HA1 increases the HIF-1α protein level by reducing PHD2-dependent proline hydroxylation. Silencing P4HA1 reduced HIF-1α levels in both

**Fig. 2** P4HA1 induces the HIF-1 pathway in breast cancer cells. **a** Gene co-expression and gene ontology analyses of the gene signature associated with P4HA1 expression in human breast cancer tissue. **b**, **c** Western blot analyses of HIF-1α expression in P4HA1-expressing MCF10A and HMLE cell lines under normoxia and hypoxia conditions. **d**, **e** HIF-1α protein levels were assessed in control and P4HA1-silenced TNBC cell lines under normoxia and hypoxia conditions. **f** HIF-1α protein levels in MDA-MB-231 cells after P4HA inhibitor 1,4-DPCA or ethyl-3,4-dihydroxybenzoic acid (DHB) treatment. **g** HRE reporter analysis of the HIF-1 activity in control and P4HA1-silenced 293FT cells under normoxia and hypoxia conditions. Results are presented as mean ± SEM; $n = 3$; **$p < 0.01$, one-way ANOVA test. **h**, **i** Quantitative RT-PCR measuring mRNA levels of HIF-1 target genes lactate dehydrogenase A (LDHA) and pyruvate dehydrogenase kinase 1 (PDK1) in control, P4HA1-silenced MDA-MB-231 cells, and P4HA1-expressing MCF10A cells under normoxia and hypoxia conditions. Results are presented as mean ± SEM; $n = 4$; *$p < 0.05$, **$p < 0.01$, independent Student's $t$ test

normoxia and hypoxia conditions (Fig. 2d, e), indicating that P4HA1-induced HIF-1α stabilization is independent of oxygen. These results suggest that the P4HA1 expression may affect the activity of PHD by changing either the concentration of the co-substrate α-KG and/or the product succinate or related metabolites (Fig. 4a). It has been reported that the PHD Km value for α-KG is about 3-fold relative to P4HA1[30]. We therefore determined whether P4HA1 reduces proline hydroxylation on HIF-1α by modulating α-KG levels in cytoplasm. We have used stable isotope resolved metabolomics (SIRM) that combines the use of stable isotope tracers with mass spectrometric (MS) and nuclear magnetic resonance (NMR) analyses[31,32], to determine the levels and $^{13}$C incorporation into various metabolites. The levels of α-KG were increased in P4HA1-silenced MDA-MB-231 cells (Fig. 4b). We further confirmed that knockdown of P4HA1 significantly increased cytoplasmic α-KG levels in MDA-MB-231, Hs578T, and MDA-MB-157 cell lines with the α-KG quantification assay (Fig. 4c). In contrast, expression of P4HA1/P4HB significantly reduced cytoplasmic α-KG levels in MCF10A cells (Fig. 4d). The hydroxylation process on collagen requires α-KG as a co-substrate, and the reaction also generates a stoichiometric amount of succinate[3]. In fact, silencing P4HA1 significantly reduced succinate levels in the cytoplasm of breast cancer cells (Fig. 4e). The changes of α-KG and succinate in P4HA1-silenced cells can be restored by P4AH1 expression (Supplementary Fig. 4a, b).

We performed two rescue experiments to determine whether α-KG and succinate mediate P4HA1-induced HIF-1α protein stabilization. Incubation of P4HA1-expressing MCF10A cells with octyl-α-ketoglutarate, a cell-permeable form of α-KG[33], reduced HIF-1α protein levels (Fig. 4f). Treatment with dimethyl-succinate slightly increased HIF-1α protein levels in P4HA1-silenced breast cancer cells (Fig. 4g). Using the ODD-luciferase reporter assay, we showed that treatment with octyl-α-ketoglutarate reduced the luciferase activity induced by P4HA1 (Fig. 4h), whereas dimethyl-succinate significantly increased luciferase activity in P4HA1-silenced cells (Fig. 4i). Consuming α-KG and generating succinate during collagen hydroxylation is dependent on the hydroxylase activity of P4HA1. We found that expressing hydroxylation-deficient P4HA1 mutants failed to induce HIF-1α at the protein levels (Fig. 4j). These results indicate that P4HA1 activates the HIF-1 pathway by reducing HIF-1α degradation, and this regulation is mediated by the metabolites α-KG and succinate.

**P4HA1 regulates cancer cell stemness through HIF-1.** Tumor-initiating cells are enriched in the CD44$^+$/CD24$^{-/low}$ cell population[34–36]. We isolated a number of CD44$^+$/CD24$^{-/low}$ and CD44$^{low}$/CD24$^+$ clones from HMLE cells[36] (Fig. 5a). P4HA1 and HIF-1α expression was upregulated in CD44$^+$/CD24$^{-/low}$ clones (Fig. 5b). The tumorsphere assay has been used to enrich tumor-initiating cells and study their colony formation activity[37]. By comparing gene expression profiles generated from tumorsphere and their matched primary human breast cancer tissue[38], we found that mRNA levels of P4HA1 were significantly increased in

tumorspheres (Fig. 5c). Importantly, CD44$^+$/CD24$^{-/low}$ cell population was decreased in P4HA1-silenced MDA-MB-231 cells (Fig. 5d). Knockdown of P4HA1 significantly inhibited tumor-sphere formation in Hs578T, MDA-MB-231, and MDA-MB-157 cell lines (Fig. 5e, f). In contrast, expression of P4HA1 and P4HB promoted mammosphere formation in MCF10A cells (Fig. 5g), while treatment with octyl-α-ketoglutarate inhibited P4HA1-induced mammosphere formation in MCF10A cells (Fig. 5g). We also found that expression of the HIF-1α PPAA mutant restored tumorsphere formation in P4HA1-silenced cancer cells (Fig. 5h). Enhanced aldehyde dehydrogenase (ALDH) activity is a marker of tumor-initiating cells[39]. Using the ALDEFLUOR™ assay and fluorescence-activated cell sorting (FACS), we showed that ALDH-high cell population was significantly reduced in P4HA1-silenced MDA-MB-231 cells, while expression of HIF-1α PPAA mutant in P4HA1-silenced cells rescued ALDH-high cell population (Fig. 5i). We also showed that expression of P4HA1 in P4HA1-silenced cells restored the tumorsphere formation and ALDH-high cell population (Supplementary Fig. 5a, b). These results suggest that activation of the P4HA1/HIF-1 pathway is crucial for stemness properties in breast cancer cells.

Reactive oxygen species (ROS) levels and oxidative phosphorylation are decreased in stem cells[40,41], and the decrease is crucial for the maintenance of stemness. We found that knockdown of P4HA1 in MDA-MB-231 cells elevated the oxygen consumption rate (OCR), indicating that oxidative phosphorylation is enhanced in P4HA1-silenced cells (Fig. 5j, k). In contrast, expression of P4HA1 and P4HB in MCF10A cells increased extracellular acidification rate (ECAR) and reduced OCR levels (Supplementary Fig. 5c, d). The upregulation of respiration in P4HA1-silenced cells was accompanied by increased ROS levels (Fig. 5l), which can be rescued by P4AH1 expression (Supplementary Fig. 5e). Expression of HIF-1α PPAA in P4HA1-silenced MDA-MB-231 cells significantly reduced ROS production and repressed oxygen consumption under both basal and maximal respiration conditions (Fig. 5k, l), indicating that P4HA1 regulates ROS and oxidative phosphorylation through the HIF-1 pathway. We also found that silence of P4HA1 reduced ECAR in breast cancer cells, and HIF-1α PPAA mutant restored ECAR in P4HA1-silenced cells (Supplementary Fig. 5f). These results suggest that upregulation of P4HA1 contributes to glycolysis in breast cancer cells. It has been shown that HIF-1 target genes LDHA and PDK1 repress oxidative phosphorylation and ROS production[42–45]. Consistently, we found that expression of the HIF-1α PPAA mutant restored the expression of LDHA and PDK1 in P4HA1-silenced cells (Supplementary Fig. 5g). Collectively, P4HA1 may regulate cancer cell stemness through HIF-1-dependent metabolic reprogramming.

**Inhibition of P4HA1 sensitizes TNBC to chemotherapy.** One crucial role for tumor-initiating cells in cancer progression is to facilitate metastatic colonization. We showed that increased P4HA1 expression in primary tumors was associated with a short distant metastasis-free survival in breast cancer patients

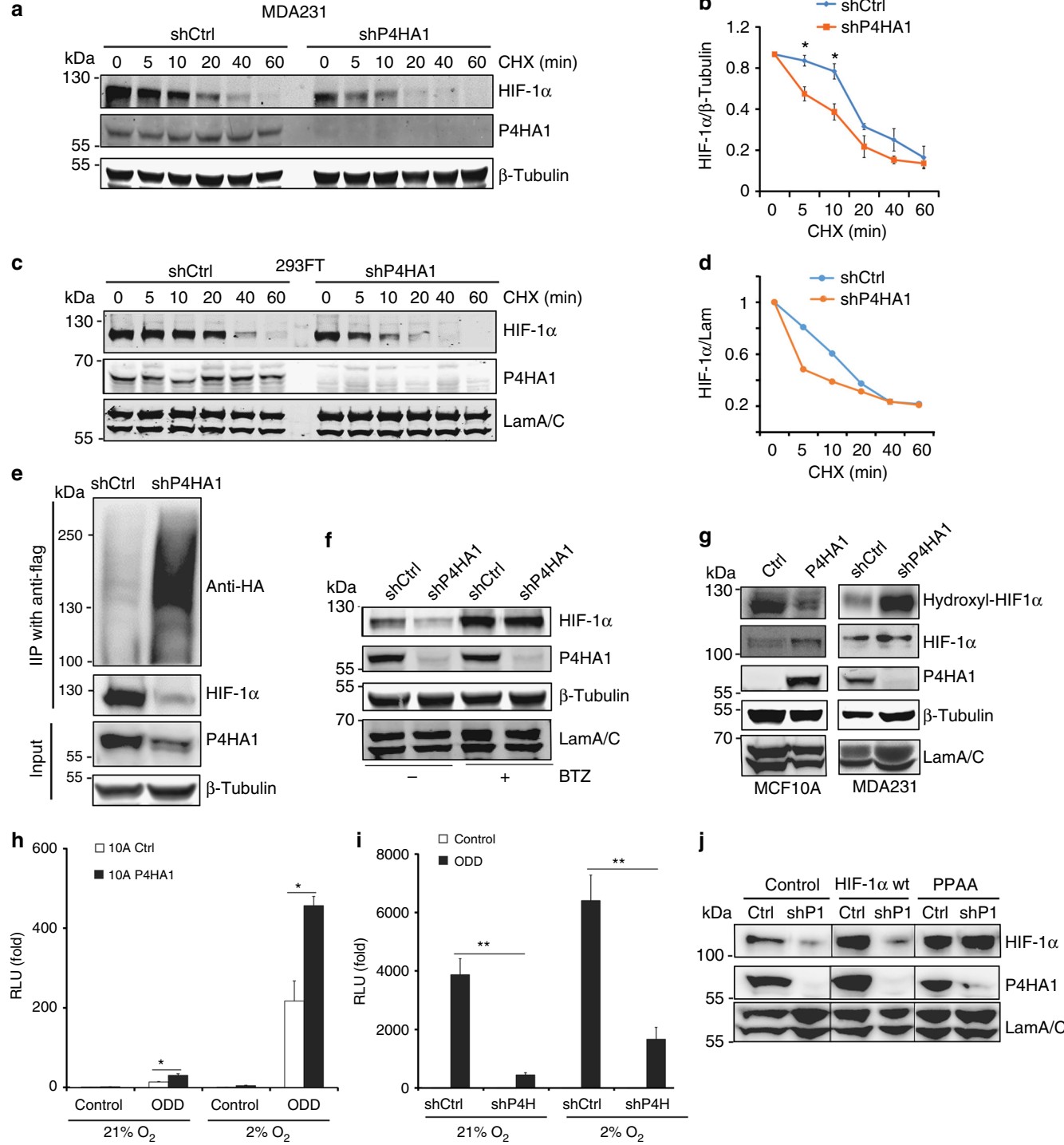

**Fig. 3** P4HA1 enhances HIF-1α protein stability. **a**, **b** Analysis of HIF-1α protein degradation in control and shP4HA1-expressing MDA-MB-231 cells with cycloheximide (CHX, 100 μM) treatment (0, 5, 10, 20, 40, and 60 min). Results are presented as mean ± SEM. $n = 3$; *$p < 0.05$, independent Student's $t$ test. **c**, **d** Analysis of HIF-1α protein degradation in control and shP4HA1-transfected 293FT cells with cycloheximide (CHX, 100 μM) treatment (0, 5, 10, 20, 40, and 60 min). **e** Immunoprecipitation and western blot analysis of HIF-1α ubiquitination in 293FT cells transfected with HIF-1α-Flag and UB-HA construct, plus shCtrl or shP4HA1 plasmids. **f** Western blot assessing HIF-1α protein levels in control and P4HA1-silenced MDA-MB-231 cells treated with proteasome inhibitor Bortezomib (BTZ) for 12 h. **g** Analysis of hydroxylated HIF-1α (P402) levels in control and P4HA1-silenced MDA-MB-231 cells, control or P4HA1-expressing MCF10A cells. **h**, **i** ODD-luciferase reporter activity in control, P4HA1-expressing MCF10A cells, and P4HA1-silenced MDA-MB-231 cells under normoxia and hypoxia conditions. Results are presented as mean ± SEM; $n = 3$; **$p < 0.01$, *$p < 0.05$, one-way ANOVA test. **j** HIF-1α protein levels were assessed in control and P4HA1-silenced MDA-MB-231 cells with wild-type HIF-1α or proline hydroxylation-deficient HIF-1α PPAA overexpression.

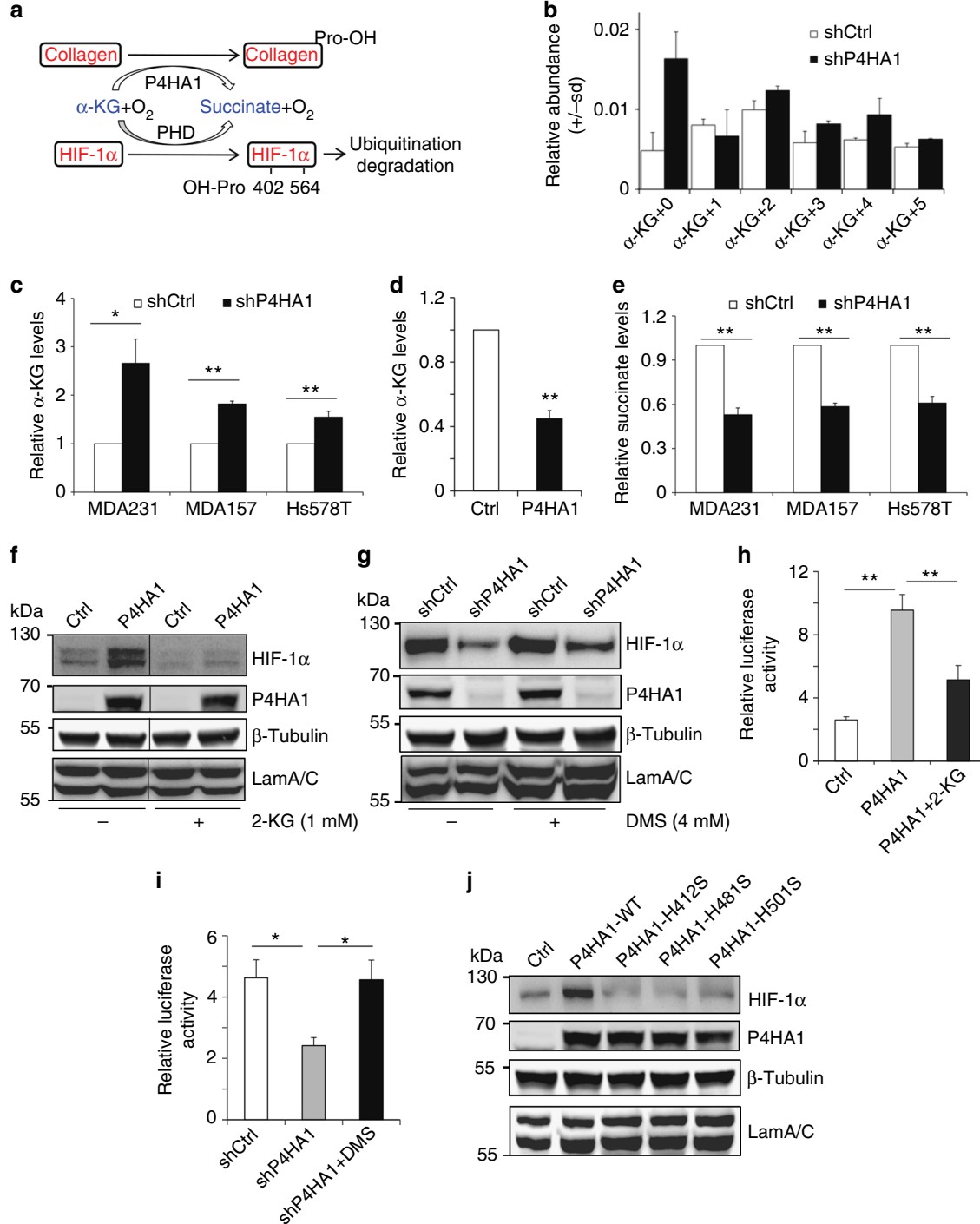

**Fig. 4** P4HA1 enhances HIF-1α stability by modulating α-KG l and succinate levels. **a** Scheme showing that HIF-1α hydroxylation is regulated by oxygen tension and metabolites (α-KG and succinate). **b–d** Cytoplasmic α-KG levels were measured in control, P4HA1-silenced MDA-MB-231, and P4HA1-expressing cells. Results are presented as mean ± SEM; $n = 3$; *$p < 0.05$, **$p < 0.01$, independent Student's $t$ test. **e** Cytoplasmic succinate levels were measured in control and P4HA1-silenced MDA-MB-231 cells. Results are presented as mean ± SEM; $n = 3$; **$p < 0.01$, independent Student's $t$ test. **f**, **g** HIF-1α protein levels were assessed in control, P4HA1-expressing MCF10A, and P4HA1-silenced cells in the presence or absence of octyl-α-ketoglutarate (1 mM) or dimethyl-succinate (4 mM). **h**, **i** ODD-luciferase reporter activity was measured in control, P4HA1-silenced, and P4HA1-expressing cells in the presence or absence of octyl-α-ketoglutarate (1 mM) or dimethyl-succinate (4 mM); $n = 3$, results are presented as mean ± SEM. *$n = 3$; $p < 0.05$, one-way ANOVA test. **j** HIF-1α protein levels were assessed in control, wild-type P4HA1-expressing MCF10A cells, and hydroxylation-deficient P4HA1 mutant-expressing MCF10A cells

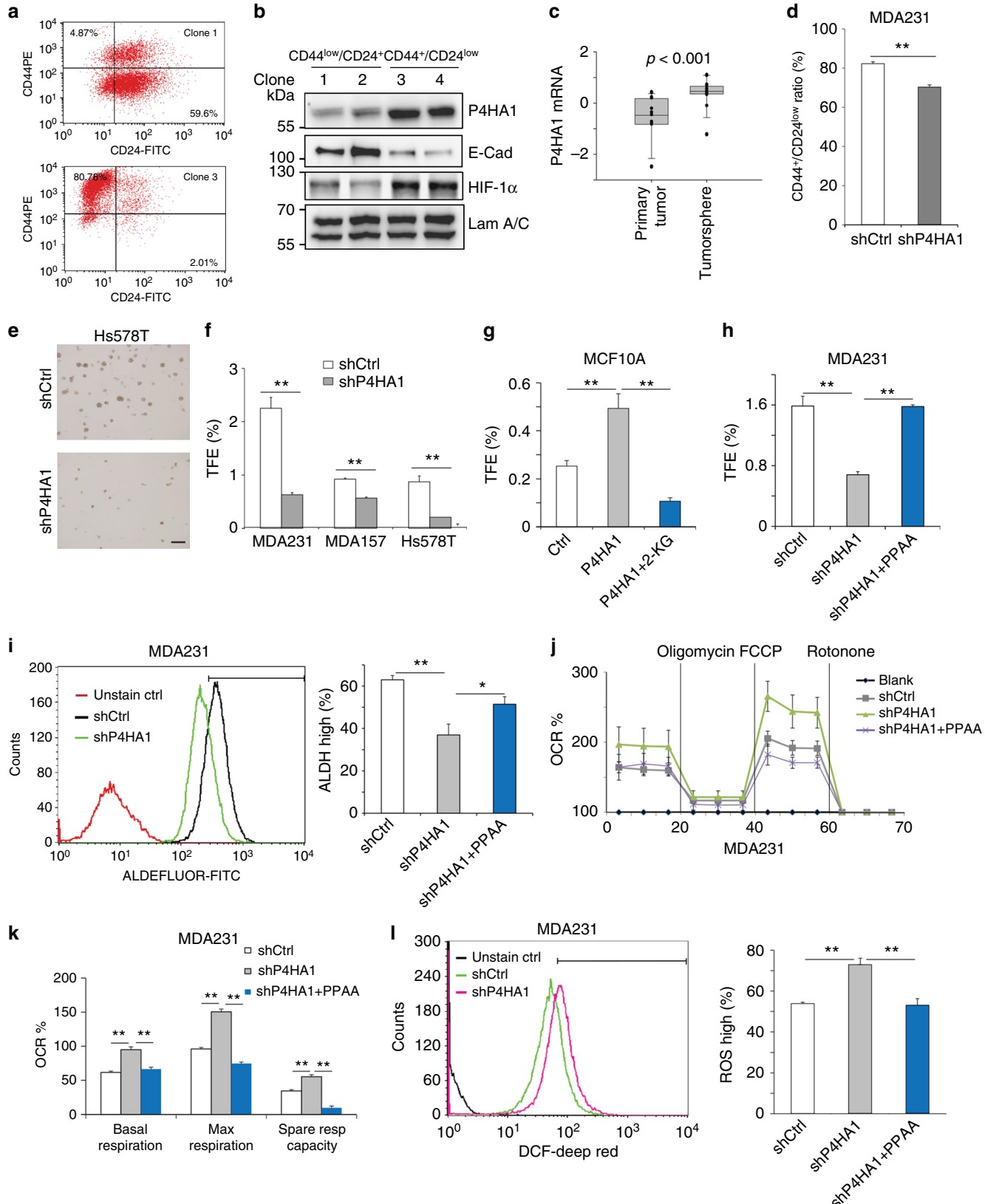

(Supplementary Fig. 6a). Therefore, we asked whether silencing P4HA1 inhibits lung colonization of MDA-MB-231 cells. Control and P4HA1-silenced MDA-MB-231 (luciferase-expressing) cells were injected in severe combined immunodeficiency (SCID) mice via tail vein. The cancer cell colonization in secondary organs was monitored by In Vivo Imaging System Facility (IVIS) imaging. We found that knockdown of P4HA1 significantly reduced colonization of MDA-MB-231 cells in lung and other organs (Fig. 6a, b) and the level of HIF-1α protein in metastasis lesions (Supplementary Fig. 6b).

**Fig. 5** P4HA regulates cancer cell stemness through the HIF-1 pathway. **a, b** Expression of HIF-1α, P4HA1, and E-cadherin was assessed in CD44[low]/CD24[+] clones and CD44[+]/CD24[-/low] clones. **c** Quantification of mRNA levels of P4HA1 in primary breast cancer tissue and tumor spheroids derived from those tissues. The gene expression values were derived from published microarray dataset. The data were log 2 transformed and mean centered. Primary tumor, $n = 11$; tumorsphere, $n = 15$. The median, 10th, 25th, 75th, and 90th percentiles were plotted as vertical boxes with error bars. The data were analyzed by Mann–Whitney rank-sum test. **d** FACS analysis of CD44[+]/CD24[-/low] population in shCtrl and shP4HA1 MDA-MB-231 cells. Results are presented as mean ± SEM; $n = 3$; **$p < 0.01$, independent Student's $t$ test. **e, f** Images of tumorspheres and quantification of tumorsphere formation efficiency (TFE) in control or P4HA1-silenced MDA-MB-231 cells, Hs578t cells, and MDA-MB-157 cells. Results are presented as mean ± SEM; $n = 3$; **$p < 0.01$, independent Student's $t$ test. **g, h** Quantification of tumorsphere formation efficiency (TFE) in control, P4HA1-expressing, and P4HA1-silenced cells in the presence or absence of octyl-α-ketoglutarate or HIF-1α PPAA. Results are presented as mean ± SEM; $n = 3$, **$p < 0.01$, one-way ANOVA test. **i** FACS analysis of aldehyde dehydrogenase (ALDH) activity in control, P4HA1-silenced MDA-MB-231 cells, and P4HA1-silenced MDA-MB-231 cells with HIF-1α PPAA. Results are presented as mean ± SEM; $n = 3$; *$p < 0.05$; **$p < 0.01$, one-way ANOVA test. **j** Seahorse analysis of oxygen consumption rate (OCR) in control, P4HA1-silenced MDA-MB-231 cells, and P4HA1-silenced MDA-MB-231 cells with HIF-1α PPAA. **k** Quantification of OCR from **j**; $n = 30$; results are presented as mean ± SEM; **$p < 0.01$, one-way ANOVA test. **l** FACS quantification of ROS levels in control or P4HA1-silenced MDA-MB-231 cells in the presence or absence of HIF-1α PPAA; $n = 3$, results are presented as mean ± SEM; **$p < 0.01$, one-way ANOVA test

The increased tumor-initiating cell population in cancer tissue has been linked with chemoresistance[26,38,46]. Consistently, we found that stem cell-enriched CD44[+]/CD24[-/low] HMLE clones were more resistant to doxorubicin (DOX) and docetaxel (DOC) treatment (Fig. 6c). These clones exhibited increased P4HA1 expression (Fig. 5a, b). More importantly, silencing P4HA1 or treatment with the P4HA inhibitor sensitized the CD44[+]/CD24[-/low] clones and MDA-MB-231 cells to DOX or DOC (Fig. 6d–f), while inhibition of P4HA1 in CD44[low]/CD24[+] clones did not significantly increase their sensitivity to DOC and DOX (Supplementary Fig. 6c). We also found that DOC or DOX treatment enhanced tumorsphere formation in MDA-MB-231 cells (Fig. 6g). Treatment with DOC or DOX for 7 days also induced P4HA1 expression (Supplementary Fig. 6d). These results suggest that chemotherapy may enrich cancer stem cells with high levels of P4HA1. Platin salt is another common chemotherapy agent for breast cancer. Silencing P4HA1 or treatment with the P4Hi also sensitized MDA-MB-231 cells to cisplatin (Supplementary Fig. 6e, f). Silencing P4HA1 or treatment with P4Hi reduced DOC-induced tumorsphere formation (Fig. 6g, h). Therefore, inhibition of P4HA1 may sensitize TNBC to chemotherapeutic agents by inhibiting cancer cell stemness.

To determine in vivo function of P4HA1 in regulating chemoresistance, we transplanted control and P4HA1-silenced MDA-MB-231 cells into mammary fat pads of female SCID mice. Treatment with DOC-induced regression in P4HA-silenced tumors, but only inhibited tumor growth in the control group (Fig. 7a). The DOC treatment was stopped after three weeks, and tumor regrowth was significantly delayed in the P4HA1-silenced group (Fig. 7a). We showed that CD44[+]CD24[-] cell population was reduced in P4HA1-silenced groups with or without DOC treatment (Supplementary Fig. 7a). These results suggest that inhibition of P4HA1 reduces the tumor-initiating cell population in vivo. Silence of P4HA1 in HIF-1α PPAA-expressing MDA-MB-231 cells did not significantly sensitize the xenograft to DOC treatment (Supplementary Fig. 7b), suggesting that P4HA1 promotes chemoresistance at least partially through the HIF-1 pathway.

Ethyl-3,4-dihydroxybenzoic acid (DHB) is a selective collagen P4Hi[30] and has been safely used in mice[7]. To determine whether inhibition of P4HA activity sensitizes TNBC to chemotherapeutic reagents, MDA-MB-231 cells were implanted in mammary fat pad, and then treated with DHB and/or DOC. Treatment with DHB only slightly inhibited tumor growth, but DHB significantly enhanced the inhibitory activity of DOC (Fig. 7b). We showed that the combined treatment induced tumor repression (Fig. 7b). We also noticed that the P4Hi was more potent to inhibit primary tumor growth. P4HA family includes P4HA1, P4HA2, and P4HA3[2]. DHB may target other P4HA family members. It is possible that P4Hi treatment suppresses tumor growth by inhibiting other P4HA family members.

Immunohistochemical (IHC) analysis data showed that DHB significantly enhanced DOC-induced cell apoptosis and growth arrest in tumor tissues, although DHB alone only slightly repressed cell proliferation and apoptosis (Fig. 7c–e). We found that the combined treatment reduced the HIF-1α protein levels and collagen deposition in tumor tissue (Fig. 7c and Supplementary Fig. 7c, d). In addition, the combined treatment blocked the metastasis of MDA-MB-231 cells to the lungs (Fig. 7f). Therefore, inhibition of P4HA1 expression or its activity is a promising strategy to sensitize the TNBC xenograft to DOC treatment.

Patient-derived tumor organoids (PDOs) and xenografts (PDXs) represent powerful tools to study drug response and drug resistance for cancer treatment because they contain stromal components and maintain the heterogeneity, differentiation status, and histopathology of the primary tumor[47,48]. We showed that TNBC PDOs grew in the 3D laminin-rich extracellular matrix (ECM) gel, and DOC or DOX inhibited PDO growth (Fig. 8a, b). Importantly, treatment with the P4HA inhibitor significantly enhanced tumor regression induced by chemotherapeutic agents (Fig. 8a, b). Similar results were also obtained in primary tumor organoids derived from other TNBC patients (Supplementary Fig. 8a, b). Using the PDX model, we further confirmed that treatment with the P4HA inhibitor enhanced the inhibitory activity of DOC to TNBC in vivo (Fig. 8c) (Supplementary Fig. 8c–d). DHB significantly increased DOC-induced cell apoptosis and growth arrest in tumor tissues (Supplementary Fig. 8e–f). The combined treatment also reduced the HIF-1α protein levels and collagen deposition in tumor tissue (Supplementary Fig. 8e–f).

To address the clinical relevance of enhanced P4HA1 expression, we assessed the association between mRNA levels of P4HA1 and patient survival using the published microarray data generated from 581 human TNBC tissue samples[49]. We found that increased P4HA1 mRNA levels correlated with short relapse-free survival in TNBC patients (Fig. 8d). A positive correlation between increased P4HA1 expression and a high incidence of recurrence was identified in TNBC patients who received chemotherapy (Fig. 8e). Using the METABRIC dataset[50], we further confirmed the association between P4HA1 expression and chemoresistance (Supplementary Fig. 8g). Next, we determined the association between P4HA1 expression and the activation of HIF-1 pathway in human breast cancer tissue by analyzing expression of the HIF-1 gene signature[51]. Expression of the HIF-1 gene signature was significantly correlated with P4HA1 levels in human breast cancer tissue (Fig. 8f). Collectively, these results suggest that activation of the P4HA1/HIF-1 axis is associated with chemoresistance and a poor clinical outcome in TNBC.

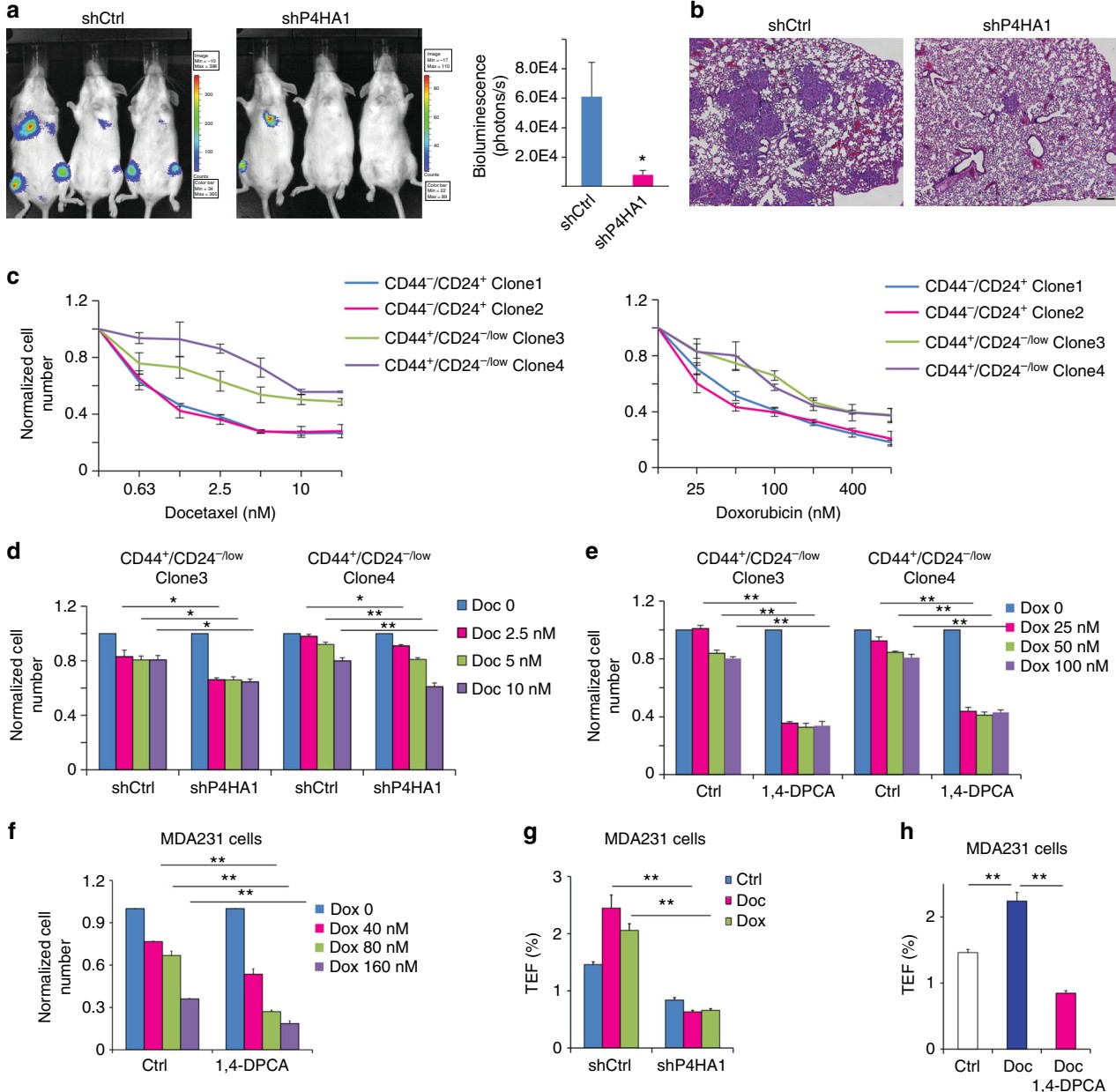

**Fig. 6** Inhibition of P4HA1 enhances cell response to chemotherapeutic agents. **a, b** Colonization of control and P4HA1-silenced MDA-MB-231 cells in the distance organs were assessed by bioluminescence signals and IHC analysis. Bar: 80 μm. Results are presented as mean ± SEM; $n = 6$; *$p < 0.05$. **c–e** Quantification of cell number in CD44[+]/CD24[-/low] HMLE clones and CD44[low]/CD24[+] HMLE clones after doxorubicin and docetaxel treatment in the presence or absence of P4HA inhibitor (1,4-DPCA) or shP4HA1. Results are presented as mean ± SEM; $n = 3$ in **c**; $n = 4$ in **d, e**; *$p < 0.05$; **$p < 0.01$, one-way ANOVA test. **f** Quantification of cell number in control and P4HA1-silenced MDA-MB-231 cells after doxorubicin treatment. Results are presented as mean ± SEM; $n = 3$; **$p < 0.01$, one-way ANOVA test. **g, h** Quantification of tumorsphere formation efficiency (TFE) in control or P4HA1-silenced MDA-MB-231 cells. The cells were treated with doxorubicin, docetaxel, and/or the 1,4-DPCA before the tumorsphere formation assay for 4 days. Results are presented as mean ± SEM; $n = 3$; **$p < 0.01$, one-way ANOVA test

## Discussion

In the present study, we show that the major role for collagen hydroxylation enzyme P4HA1 in breast cancer cells is to enhance HIF-1 protein stability by reducing availability of α-KG and by producing succinate. Inhibition of the P4HA1/HIF-1 axis suppresses stemness in breast cancer cells. We demonstrate that P4HA1 expression is upregulated in TNBC tissues and correlates with poor prognosis and chemoresistance in TNBC patients. Importantly, inhibition of collagen P4H sensitizes TNBC to chemotherapy. P4HA1 expression is also upregulated in HER2-positive tumor and cancer cell line (Supplementary Fig. 9a, b).

However, roles of P4HA1 in HER2 cancer progression still need further clarification. Our findings reveal a novel function of P4HA1 in promoting TNBC progression through the HIF-1 pathway.

PHD-dependent HIF-1α hydroxylation is enhanced by substrate α-KG and inhibited by succinate[3,33,52,53]. Mutation of IDH-1 in glioma induces the HIF-1 pathway by reducing α-KG production[53]. This evidence suggests that metabolites such as α-KG and succinate also contribute to HIF-1 activation during cancer development and progression. However, how those metabolites are regulated in breast cancer to induce hyperactivation of the

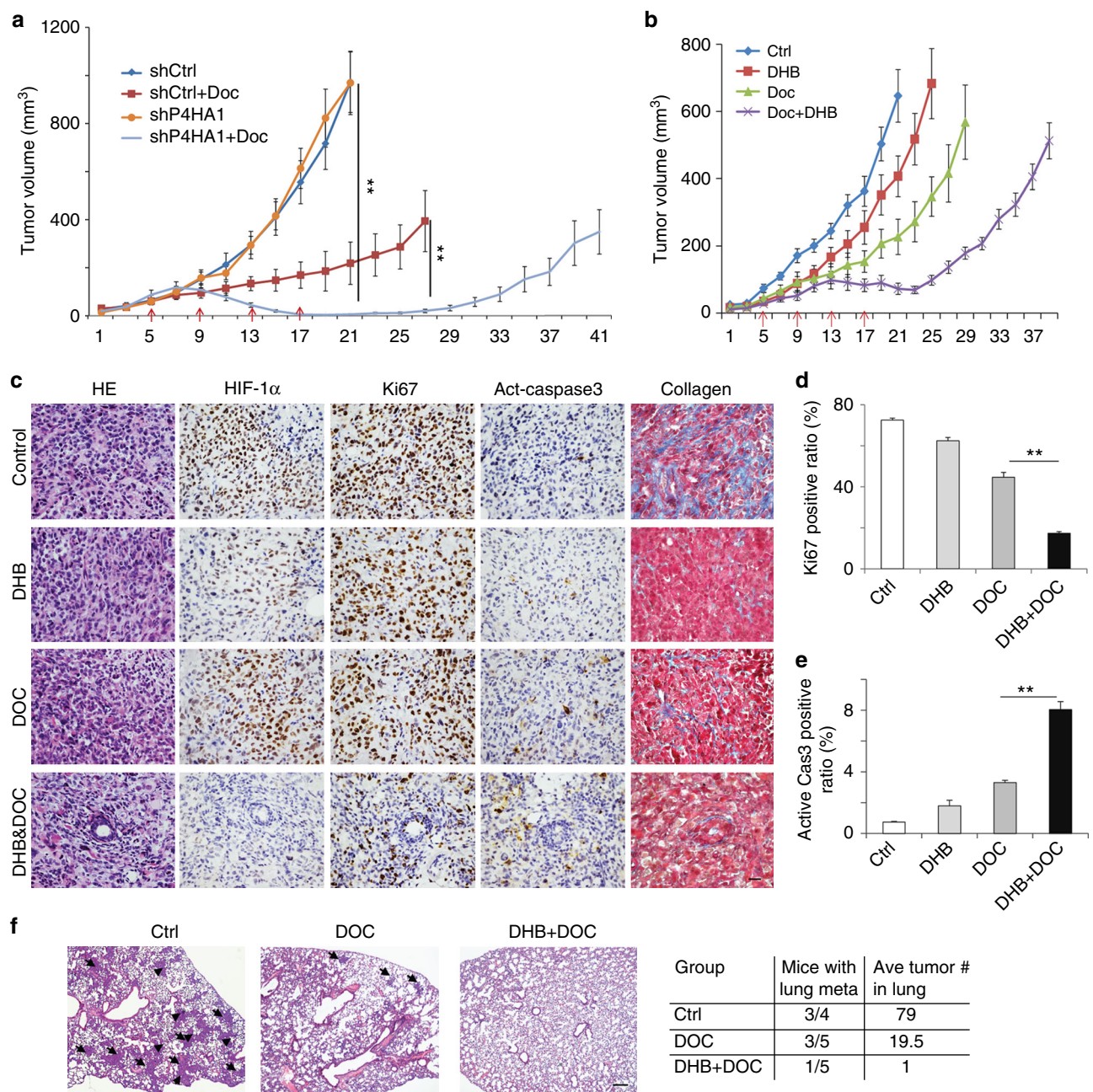

**Fig. 7** Inhibition of P4HA1 sensitizes TNBC to docetaxel in vivo. **a** Tumor growth curve of control and P4HA1-silenced MDA-MB-231 cells implanted mice treated with docetaxel. Results are presented as mean ± SEM; $n = 10$; **$p < 0.01$. The data were analyzed by one-way ANOVA test. **b** Tumor growth curve of MDA-MB-231 cells implanted mice treated with P4H inhibitor DHB (40 mg/kg) and/or Doc (docetaxel) (10 mg/kg). The DHB treatment started from day 1 to day 21 in the tumor growth curve. Results are presented as mean ± SEM; $n = 8$. **c** HE images, IHC staining images (HIF-1α, Ki-67, active-caspase-3), and Masson's trichrome staining images of tumor sections from mice treated with DHB and/or Doc. Bar: 20 μm. **d**, **e** Quantification of cell proliferation and apoptosis in tumor sections from mice treated with DHB and/or Doc; $n = 4$; **$p < 0.01$, one-way ANOVA test. **f** Representative HE images and quantification data of lung metastasis in mice treated with DHB and/or Doc. Bar: 80 μm

HIF-1 pathway is unclear. Collagen P4H has 3-fold higher affinity to α-KG and 6-fold higher affinity to oxygen compared to PHD[30]. About 38% of proline in collagen are hydroxylated[1], thus increased expression of collagen P4H may consume a substantial amount of α-KG and generates succinate. We showed that P4HA1 protein expression is significantly upregulated in TNBC, and that the upregulation is associated with TNBC progression. Importantly, P4HA1 expression significantly reduced α-KG levels and elevated succinate levels in the cytoplasm, and subsequently enhanced HIF-1α stability and activity. These findings do not rule

out the possibility that other pathways may also contribute to the differential activation of the HIF-1 pathway between breast cancer subtypes. It has been shown that XBP1 assembles a transcriptional complex with HIF-1α that regulates the expression of HIF-1α-target genes and subsequently drives TNBC tumorigenicity[19]. In addition, SHARP1 binds to HIFs and promotes proteasomal degradation of HIFs by serving as the HIF-presenting factor to the proteasome in breast cancer progression[54]. Nevertheless, the fact that P4HA1 expression correlates with HIF-1 activation in breast cancer tissues suggests that

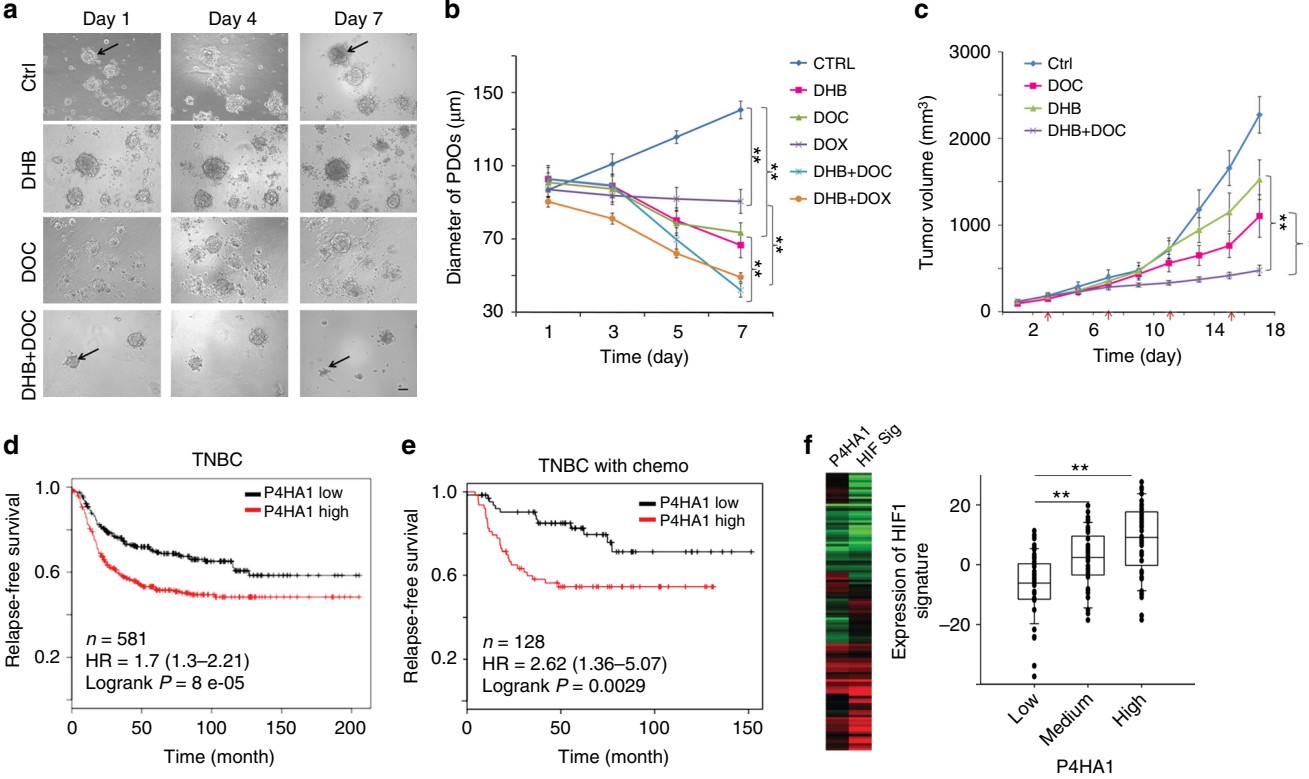

**Fig. 8** P4HA1 expression is associated with chemoresistance and activation of the HIF-1 gene signature. **a**, **b** Growth of patient-derived tumor organoids (PDOs) in the 3D laminin-rich ECM gel was assessed after docetaxel (DOC), doxorubicin (DOX), and/or P4HA inhibitor (DHB) treatment; results are presented as mean ± SEM. $n = 89$; **$p < 0.01$, one-way ANOVA test. Bar, 40 μm. **c** Tumor growth curve of PDX treated with DHB (40 mg/kg) and/or DOC (10 mg/kg); $n = 5$; **$p < 0.01$; *$p < 0.05$, one-way ANOVA test. **d** The association of P4HA1 expression with recurrence-free survival was assessed by analyzing the mRNA levels in basal-like/TNBC tissues, $n = 580$, Kaplan–Meier survival analysis. **e** The association of P4HA1 expression with recurrence-free survival was assessed by analyzing the mRNA levels in basal-like/TNBC tissues from patient received adjuvant and neoadjuvant chemotherapy, $n = 128$, Kaplan–Meier survival analysis. **f** Heatmap and Box plot showed the association of P4HA1 levels and expression of the HIF-1 gene signature in human breast cancer tissue. The median, 10th, 25th, 75th, and 90th percentiles were plotted as vertical boxes with error bars, $n = 39$, **$p < 0.01$, one-way ANOVA test

hyperactivation of HIF-1α in TNBC is at least partially induced by P4HA1.

Tumor-initiating cells have the ability to form tumors after xenotransplantation and are the driver of cancer progression[55,56]. We showed that P4HA1 expression was upregulated in tumor-initiating cell-enriched populations. P4HA1 expression promoted tumorsphere formation, expanded the ALDH-high cell population, and enhanced cancer cell colonization in the distant organs. These results suggest that upregulation of P4HA1 contributes to the survival or self-renewal of tumor-initiating cells. Self-renewal of stem cells relies primarily on glycolysis and the pentose phosphate pathway[57]. This process is also associated with a reduction of oxidative phosphorylation. We found that P4HA1 expression inhibited oxidative phosphorylation and reduced ROS levels in breast cancer cells. The majority of ROS are generated in mitochondria during oxidative phosphorylation. ROS directly react with various proteins, such as kinases, phosphatases, or transcription factors. Reduced ROS levels are critical for survival and self-renewal of stem cells;[58] therefore, ROS are considered as crucial mediators of metabolism and stem cell fate decisions. HIF-1 has been identified as a major metabolic regulator in stem cells[57], and activation of the HIF-1 pathway induces glycolysis and suppresses oxidative phosphorylation[59,60]. We found that P4HA1 expression reduced oxidative phosphorylation and induced HIF-1-targeted gene expression in breast cancer cells. The rescue experiments further demonstrate that P4HA1 inhibited ROS generation and oxidative phosphorylation through the

HIF-1 pathway. These results identified a pathway that collagen P4H modulates cell metabolism and subsequently enhances tumor-initiating cell activity.

Chemotherapy is widely used for TNBC treatment. It has been shown that treatment with chemotherapeutic agents increases the tumor-initiating cell population in cancer tissue[26,46]. The tumor-initiating cells with enhanced ALDH expression and ATP-binding cassette transporter activity further promote the development of chemoresistance[61]. Thus, targeting tumor-initiating cells is a promising approach to overcome chemoresistance[55]. Silence or inhibition of P4HA1 activity reduced ALDH-high cell population and enhanced inhibitory activity of DOC and DOX in CD44[+]/CD24[−/low] cells. We also demonstrate that combined treatment with the P4Hi and DOC induces regression of TNBC in PDOs and orthotopic breast cancer models. Moreover, increased P4HA1 expression correlates with poor prognosis and chemoresistance in TNBC patients. These results suggest that collagen P4HA1 is a potential drug target and prognosis marker for TNBC. Therefore, developing highly specific and potent collagen P4Hi may hold the key to overcome chemoresistance and improve TNBC treatment. Roles of signal transducer and activator of transcription 3 (STAT3) in chemoresistance are well established[62,63]. However, the unbiased gene co-expression analysis did not detect the association between P4HA1 and the STAT3 pathway or gene signature in human breast cancer tissues. Currently we do not have strong data to support that P4HA1 promotes chemoresistance through the STAT3 pathway.

P4HA1 expression is induced during breast cancer development and TNBC progression. However, it is unclear how P4HA1 expression is induced in TNBC. P4HA1 expression is induced by HIF-1α under hypoxia condition[64]. Activation of the HIF-1 pathway enhances collagen production and tissue fibrosis. These results suggest a positive feedback loop between collagen hydroxylation and the HIF-1 pathway, and this feedback loop may be critical for the integration of collagen deposition and TNBC progression. Interestingly, silencing HIF-1α partially inhibited P4HA1 expression in TNBC cells[7], suggesting that HIF-1 is not the only pathway that regulates P4HA1 expression during breast cancer progression. TNBC is associated with activation of the epithelial–mesenchymal transition (EMT) program[65]. It is plausible that P4HA1 expression is regulated by EMT drivers during breast cancer progression.

In the present study, we have identified a novel function of collagen P4H in regulating the HIF-1 pathway and tumor-initiating cells in TNBC. These findings fill a critical gap in our knowledge about the HIF-1 regulation and chemoresistance during TNBC progression. We propose that targeting collagen P4H provides a potential strategy to improve TNBC treatment.

## Methods

All the reagents used in this study were listed in Supplementary Table 2

**Quantitative RT-PCR**. Total RNA was extracted from cells using TRIzol reagent (Invitrogen, 15-596-018). Complementary DNA was synthesized using the SuperScript™ III First-Strand Synthesis System (Invitrogen, 18-080-051) from 1.0 μg RNA samples. Quantitative RT-PCRs were carried out with SYBR Green PCR Master Mix reagents using an StepOnePlus™ Real-Time PCR System (Thermo Fisher Scientific, 4376600). Thermal cycling was conducted for 30 s at 95 °C, followed by 40 cycles of 5 s amplification at 95 °C, 30 s at 55 °C, and 15 s at 72 °C. The relative quantification of gene expression for each sample was analyzed by the threshold cycle (CT) method. Information of primers used for amplification of PDK1, LDHA, and 18S ribosomal RNA is given in Supplementary Table 3.

**Western blot assays**. Cells were lysed in 2% sodium dodecyl sulfate (SDS) in phosphate-buffered saline (PBS) buffer containing phosphatase and protease inhibitor cocktails (EMD Millipore, 539131). Protein concentration was measured using Pierce™ BCA Protein Assay Kit (Thermo Fisher Scientific, 23227). Equal amounts of protein lysates or cell conditional medium (normalized to cell numbers) were subjected to SDS gel electrophoresis, immunoblotted with primary antibodies, and horseradish peroxidase (HRP)-conjugated or DyLight 680/800-conjugated secondary antibodies. Western blot results were quantified by using the AlphaInnotech analysis software. The secondary antibodies used in this study were as follows: HRP-conjugated goat anti-rabbit IgG secondary antibody (Thermo Fisher Scientific, 31460), HRP-conjugated goat anti-mouse IgG secondary antibody (Thermo Fisher Scientific, 31440), HRP-conjugated rabbit anti-goat IgG secondary antibody (Thermo Fisher Scientific, 31402); DyLight 680-conjugated goat anti-rabbit IgG secondary antibody (Thermo Fisher Scientific, 35569), DyLight 800-conjugated goat anti-mouse IgG secondary antibody (Thermo Fisher Scientific, SA5-35521), DyLight 680-conjugated donkey anti-goat IgG secondary antibody (Thermo Fisher Scientific, SA5-10090). The primary antibodies used in this study were listed in Supplementary Table 4. Uncropped scans of important blots are included in the Supplementary Figure 10–15.

**Luciferase reporter assays**. Luciferase reporter construct containing three HREs was purchased from Addgene (26731). HRE-luciferase reporter plasmid and Renilla luciferace vector were transiently transfected into control and P4HA1-silenced MDA-MB 231 cells. Cell lysate was collected for the luciferase assay analysis 48 h after transfection.

The ODD-luciferase reporter construct consisting of a firefly luciferase gene fused to the hydroxylation-dependent degradation region of HIF-1α was purchased from Addgene (18965). The ODD-luciferase reporter plasmid and Renilla luciferace vector were transiently transfected into control and P4HA1-silenced MDA-MB-231 cells, or P4HA1-expressing and P4HB-expressing MCF10A cells. For rescue experiments, the ODD-luciferase reporter plasmid and Renilla luciferace vector were transiently transfected into P4HA1-silenced MDA-MB-231 cells followed by succinate treatment or P4HA1-expressing and P4HB-expressing-MCF10A cells followed by α-KG treatment.

**Immunoprecipitation**. HIF-1α expression plasmid pCDH1-HIF-1α(wild-type)-Flag, UB-HA plasmid, and shctrl-transfected or shP4HA1-transfected 293FT cells were lysed with ice-cold hypotonic gentle lysis buffer (10 mM Tris-HCl (pH 7.5),

10 mM NaCl, 2 mM EDTA, 0.5% Triton X-100, phosphatase and protease inhibitor cocktails (EMD Millipore, 539131)) and incubated on ice for 10 min. Five moles of NaCl was added to the cell lysis buffer to a concentration of 150 mM. The protein complexes were pulled down with anti-Flag M2 affinity gel (Sigma-Aldrich, A2220), and the immunoprecipitated protein was eluted for western blot analysis.

**Stable isotope tracers experiment**. When cells in 10 cm plates were about 70% confluent in the cell growth medium, the cell growth medium was changed to stable isotope tracer medium containing [U-$^{13}$C]-glucose (CLM-1396-PK) at 0.45%, 10% dialyzed fetal bovine serum, and 1× penicillin–streptomycin in glucose-free Dulbecco's modified Eagle's medium (DMEM)). The cells were incubated in a tracer medium with 5% $CO_2$ at 37 °C and recorded the start time. At the end of the tracer treatment (24 h later), the cell culture plates were placed on ice, the medium was removed, and the plates were washed three times with ice-cold PBS. One milliliter cold $CH_3CN$, 0.55 ml $H_2O$, and 0.2 ml 0.2 mM Tris-HCl (pH 8.0) were added into each plate and cells were scraped and collected in 15 ml tubes. One milliliter cold chloroform was added in the 15 ml tube containing cell samples. After shaking 60 times and vortexed, the 15 ml tube containing cell samples was centrifuged at 3500 × g for 20 min at 4 °C. The top layer (polar extraction) was transferred to the pre-weight 15 tubes with fine tip pipette. Exact weight for the polar extraction was recorded and two aliquots of 1/8 of the total polar extract volume were transferred to 1.5 ml glass vials for gas chromatography-mass spectrometry analysis. Two aliquots of 1/16 of the total polar extract volume were transferred to 0.5 ml screw top microfuge tubes for Fourier transform-mass spectrometry analysis. The remaining polar extract was aliquoted into two 2.0 ml microfuge tubes for NMR analysis. The middle layer (protein) was transferred to pre-weight 1.5 ml Eppendorf tube and washed with 500 μl cold methanol. After centrifugation at 18,407 × g for 20 min at 4 °C, the supernatant was discarded and the protein was dried using the SpeedVac (Vacufuge Plus, Eppendorf). The dry protein was weighted and store at −80 °C.

**Seahorse assay**. MDA-MB-231 cells (shcontrol, shP4HA1, or shP4HA1 plus HIF-1α PPAA) and MCF10A cells (control or P4HA1, P4HB expressing) were seeding to a 96-well Seahorse XF Cell Culture Microplate (10,000 cells per well). After 24 h, the cell culture growth medium was changed to a pre-warmed assay medium and the cell culture microplate was placed into a 37 °C non-$CO_2$ incubator for 45 min to 1 h prior to the assay. Then, the mitochondrial respiration (OCR) and glycolysis (ECAR) were measured in the XF Analyzer according to the manual of Seahorse XF Cell Mito Stress Test Assay (Seahorse Bioscience of Agilent Technologies).

**α-KG quantification assay**. According to the kit instructions (Abcam, ab83431), MDA-MB-231 cells (shcontrol or shP4HA1) and MCF10A cells (control or P4HA1, P4HB expressing) were washed with cold PBS and resuspended in 500 μl ice-cold α-KG Assay Buffer on ice. Cells were homogenized quickly by pipetting up and down a few times and then centrifuged for 2–5 min at 4 °C at 15,871 × g to remove any insoluble material. Fifty microliters of α-KG standard or cell sample was added to the well of 96-well plates. For each α-KG standard or cell sample, the reaction mix (50 μl) was added and incubated for 30 min at 37 °C and protected from light. The 96-well plates were measured on Spectra MRTM microplate spectrophotometer (Dynex Technologies) at OD570 nm.

**Succinate quantification assay**. According to the kit instructions (Megazyme, K-SUCC), MDA-MB-231 cells (shcontrol or shP4HA1) were washed with cold PBS and resuspended in 300 μl ice-cold PBS. Sonicated for several seconds, 10 μl sample or succinate standard was added to the well of 96-well plates. For each α-KG standard or cell sample, a reaction mix (200 μl distilled water, 20 μl buffer, 20 μl NADH, 20 μl ATP/PEP/CoA, 2 μl PK/L-LDH) was added and the absorbance ($A1$) was read at OD340 nm after 3 min. After adding 2 μl succinyl-CoA synthetase, the absorbance ($A2$) was read on a Spectra MRTM microplate spectrophotometer (Dynex Technologies) at OD340 nm at the end of the reaction (6 min later). The concentrations of succinate in samples were calculated by the $\Delta A$ method.

**Cancer stem cell ALDH FACS analysis**. According to the manual of ALDH ALDEFLUOR™ Kit (Stemcell Technologies, 1700), cells were trypsinized and resuspended in the ALDEFLUOR™ Assay Buffer at 1 × 10⁶ cells/ml buffer. Five microliters of the activated ALDEFLUOR™ reagent was added to and mixed well with 1.0 ml cell sample ("Test" tube). Then, 0.5 ml of the mixture was immediately transferred to a "control" tube by adding 5 μl of ALDEFLUOR™ DEAB reagent. "Test" and "Control" samples were incubated for 45 min at 37 °C. After incubation, samples were centrifuged and resuspended in 0.5 ml of ALDEFLUOR™ Assay Buffer and kept on ice for FACS analysis. FACS analysis was done with Becton Dickinson LSR II and data were analyzed by CellQuest Pro. Gates were set according to unstained control and single-color control for fluorescein iso-thiocyanate (FITC).

**CD44/CD24 FACS analysis**. To selected HMLE monoclones with mesenchymal traits and in the expression of stem cell markers (CD44$^+$/CD24$^{-/low}$), 1000 HMLE cells were passaged to a 100-mm culture dish. After the monocell clones' growth,

cell clones with epithelial or mesenchymal morphology were picked and expanded. The CD44/CD24 levels in these clones were examined by FACS analysis. HMLE cells were trypsinized and resuspended ($1 \times 10^6$ cells) in 100 µl HBSS buffer (0.137 M NaCl, 5.4 mM KCl, 0.25 mM $Na_2HPO_4$, 0.1% glucose, 0.44 mM $KH_2PO_4$, 1.3 mM $CaCl_2$, 1.0 mM $MgSO_4$, 4.2 mM $NaHCO_3$) containing 20 µl FITC mouse anti-human CD24 (BD Biosciences, 555427). The mixture was incubated at room temperature for 1 h and protected from light. Then, 10 µl phycoerythrin (PE) mouse anti-human CD44 (BD Biosciences, 550989) was added to each sample and incubated on ice for 20 min and protected from light. After incubation with antibodies, samples were fixed with 2% paraformaldehyde (PFA) for 20 min at room temperature. Samples were centrifuged and resuspended in 200 µl HBSS buffer and kept on ice for FACS. FACS analysis was done with Becton Dickinson LSR II and data were analyzed by CellQuest Pro. Gates were set according to unstained control and single-color controls for FITC, PE, respectively.

**ROS FACS analysis**. According to the manual of CellROX Deep Red Flow Cytometry Assay Kit (Life Technologies, C10491), cells were trypsinized and resuspended in a complete growth medium at a concentration of $0.5 \times 10^5$ cells/ml. The CellROX® Deep Red reagent was added to the cell samples at a final concentration of 500–1000 nM and incubated for 45 min at 37 °C and protected from light. One microliter propidium iodide (1 mg/ml) was added to the cell samples and incubated on ice for FACS analysis. FACS analysis was done with Becton Dickinson LSR II and data were analyzed by CellQuest Pro. Gates were set according to unstained control and single-color control for deep red.

**Tumorsphere assay**. Tissue culture plates were pretreated with poly (2-hydroxyethyl methacrylate) (12 mg/ml in 95% ethanol) overnight at 53 °C. Cells were cultured on 2D plastic culture dish in a regular culture medium and trypsinized according to the standard protocol. Cells were filtered by Cell Strainer (40 µm, Corning®) to ensure a single-cell suspension. Viable cell number was calculated after trypan blue staining using a hemocytometer. One milliliter of tumorsphere media (DMEM/F12 medium supplemented with B27 (1:50), epidermal growth factor (20 ng/ml), basic fibroblast growth factor (20 ng/ml), insulin (5 µg/ml), hydrocortisone (0.5 µg/ml), gentamycin (100 µg/ml)) were added into each well of a non-adherent 12-well culture plate. Single-cell suspension was plated at an appropriate density in triplicate. After incubating for 5 days in a humidified atmosphere with 5% $CO_2$ at 37 °C without moving or disturbing the plates, the number of tumorspheres >50 µm diameter was counted using a microscope. Tumorsphere-forming efficiency (%) is calculated as follows: number of tumorspheres per well/number of cells seeded per well × 100.

**Cell proliferation assay**. MDA-MB-231 cells or HMLE cells were seeded to a 96-well cell culture plate (3000 cells per well). After 24 h, P4Hi or combination with DOC, cisplatin was added to the cell culture medium. After a 2-day treatment, 10 µl of the CCK-8 solution was added to each well of the plate, according to the instruction of the Cell Counting Kit-8 (Sigma-Aldrich, 96992). The plate was incubated for 1–4 h at 37 °C in the cell incubator . The 96-well plate was measured on Spectra MRTM microplate spectrophotometer (Dynex Technologies) at OD450 nm. For each cell clone, the OD450 values in the drug treatment group were normalized to the value in the control group.

**In Vivo xenograft experiments**. Six-week-old female SCID mice were randomly grouped and injected with $1 \times 10^6$ shcontrol or shP4HA1 MDA-MB-231/Luc cells in mammary fat pads. Mice were treated with DOC (10 mg/kg) every 4 days by intraperitoneal injection when tumors reached 60–70 $mm^3$ . Tumors were measured with a caliper every 2 days to analyze tumor growth. At the experimental endpoint, tumors and lung tissues were harvested and fixed with 4% PFA for paraffin-embedded section.

Six-week-old female SCID mice were injected with $1 \times 10^6$ MDA-MB-231 cells in mammary fat pads and randomly grouped into four groups: group 1, control; group 2, P4H inhibitor (P4Hi); group 3, DOC; group 4, P4Hi and DOC. DHB has been safely used in vivo to inhibit collagen deposition at 40 mg/kg weight per day[7]. It also inhibits cancer metastasis at this concentration[7]. Thus, we chose this inhibitor for the xenograft experiments. DHB (40 mg/kg) was administered every day by intraperitoneal injection, 2 weeks after tumor cell injection. Mice were treated with DOC (10 mg/kg) every 4 days by intraperitoneal injection when tumors reached 60–70 $mm^3$. Tumors were measured with a caliper every 2 days to analyze tumor growth. After treatment with four doses of DOC (day 21), the tumors were kept in mice to examine the tumor regrowth without drug treatment. To examine the effects of drug treatments on cell proliferation and apoptosis, we performed surgery to remove tumors at day 21 for hematoxylin and eosin (H&E) and IHC analyses. The mice were kept for another month after the surgery to detect metastasis to lung. At the experimental endpoint, lung tissues were harvested and fixed with 4% PFA for H&E analyses.

Six-week-old female SCID mice were randomly grouped and injected with $1 \times 10^6$ (in 200 µl PBS) shcontrol or shP4HA1 MDA-MB-231/Luc cells via the tail vein. To detect lung metastasis, bioluminescent images were taken 2 weeks after cancer cell injection with the IVIS Spectrum. Mice were sacrificed 5 weeks after cancer cell injection.

All procedures were performed within the guidelines of the Division of Laboratory Animal Resources at the University of Kentucky.

**Human breast cancer PDO experiments**. Human breast cancer tissues were minced with scalpels until finely chopped, and then were transferred to a 50 ml conical vial using a cell lifter. The minced tumor tissues were incubated with 1 mg/ml type IV collagenase (Sigma #C5138) and 100 U/ml hyaluronidase (Sigma #H3884) in DMEM/F12 at 37 °C for 2 h. Digested tumor fragments were pelleted for 3 min at $100 \times g$, and the supernatant was discarded. Large tumor fragments were removed by 100 µm filter. Isolated primary tumor organoids were cultured on top of a 3D laminin-rich ECM. In general, primary tumor organoids were seeded on top of a thin gel of Engelbreth–Holm–Swarm tumor extract (Matrigel; BD Biosciences), and a medium containing 5% Matrigel was added. The P4Hi and/or chemotherapeutic agents were added after 24 h. The diameter of tumor organoids was measured under the microscope every day for the next 8 days.

**Human breast cancer PDX experiments**. Human breast cancer tissues were minced with scalpels until finely chopped, and then were mixed with Matrigel. The mixture of minced tumor tissue and Matrigel was injected subcutaneously to the NOD-SCID IL2Rgammanull female mice (Jackson Laboratory) (F1). When the tumor diameter reached ~200 mm, the tumor was collected and minced with scalpels until finely chopped, and then were mixed with Matrigel.

The mixture of minced human breast cancer tissue and Matrigel was injected subcutaneously to the 20 NOD-SCID IL2Rgammanull female mice (Jackson Laboratory) (F2). Tumors were measured with a caliper every 2 days to analyze tumor growth. All mice were randomly grouped into four groups: group 1, control; group 2, P4Hi (DHB); group 3, DOC; group 4, P4Hi (DHB) and DOC. P4Hi (DHB, 40 mg/kg) was administered every day by intraperitoneal injection, 2 weeks after tumor cell injection. Mice were treated with DOC (10 mg/kg) every 4 days by intraperitoneal injection when tumors reached ~100 $mm^3$. At the experimental endpoint, tumors and lung tissues were harvested and fixed with 4% PFA for paraffin-embedded section.

PDX and PDO experiments involved human breast cancer patients. The studies were approved by the Committee on the Use of Humans as Experimental Subjects of University of Kentucky, IRB protocol # 15–0767-P6H. All clinical samples were collected with written informed consent from patients in the Markey Cancer Center at the University of Kentucky. Cell culture, Immunohistochemistry and Masson's trichrome staining are included in the Supplementary Methods.

**Patient survival analysis and other statistical analysis**. To address the clinical relevance of enhanced P4HA1 expression, we assessed the association between mRNA levels of P4HA1 and patient survival using the published microarray data generated from 580 human TNBC tissue samples[49]. Tumor samples were equally grouped into low and high P4HA1 expression based on the mRNA levels. Significant differences in overall survival time were assessed with the Kaplan–Meier survival analysis and the Cox proportional hazard (log-rank) test. The activation of the HIF-1 gene signature was evaluated based on the expression of 58 HIF-1 target genes[51]. The correlation between P4HA1 expression and activation of the HIF-1 gene signature was analyzed using the microarray dataset generated from human breast cancer tissues[66].

All experiments were repeated at least twice. Results were reported as mean ± S. E.M.; the significance of difference was assessed by $\chi^2$, independent Student's t test, or one-way analysis of variance (ANOVA) with SigmaPlot 12.3 (Systat Software Inc., San Jose, CA, USA). $P < 0.05$ represents statistical significance and $P < 0.01$ represents sufficiently statistical significance. All reported $P$ values were from two-tailed tests.

## Data availability

Metabolic data are available to the community via the Metabolomics Workbench (http://www.metabolomicsworkbench.org) under project ID PR000703 (https://doi.org/10.21228/m8t10p). All the other data generated in this study are available within the article and its Supplementary Information Files and from the corresponding author upon reasonable request.

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

## Acknowledgements

We acknowledge the assistance of the following Markey Cancer Center Shared Resource Facilities, all of which are supported by the grant P30 CA177558: the Biospecimen and Tissue Procurement Shared Resource Facility for assistance in tissue fixation and section; the Flow Cytometry and Cell Sorting Core Facility for performing FACS analysis; and the Redox Metabolism Shared Resource Facility for performing Seahorse experiments. Additionally, we thank the cancer center's Research Communications Office for assistance with manuscript preparation. This study was supported by start-up funding from Markey Cancer Center and funding support from NCI (1R01CA207772, 1R01CA215095, and 1R21CA209045 to R.X.), Markey Cancer Center CCSG pilot funding (P30 CA177558), and United States Department of Defense (W81XWH-15-1-0052 to R.X.).

## Author contributions

R.X. conceived the work, performed and supervised the data analysis, and wrote the manuscript. G.X. performed the experiments, collected and analyzed the data, and wrote the manuscript. R.L.S. carried out some experiments and analyzed the data. J.C. carried out some experiments and collected the data. T.G. provided material and revised the manuscript. T.L.S. carried out SIRM experiment and analyzed the data. L.M.S. analyzed the data and revised the manuscript. K.O.C. provided material and revised the manuscript. A.N.L. designed and analyzed the SIRM experiments, and edited the manuscript.

## Additional information

**Competing interests:** The authors declare no competing interests.

