## [Peer Review File · Nature Communications]

Reviewers' Comments:

Reviewer #1:

Remarks to the Author:

In the present study Xiong et al. describe a new mechanism contributing to HIF-1 α stability in TNBC. The authors demonstrate that expression of the collagen prolyl 4-hydroxylase subunit 1 (P4HA1) in cancer cells is essential for HIF-1 α stabilization in an oxygen independent manner. The role of HIF-1 α in breast cancer progression and metastasis has been described before, however the authors shed light on a new mechanism that further contributes to HIF-1 α mediated malignancy in TNBC. Furthermore, an interesting role of the P4HA1-HIF-1 α axis in the resistance to chemotherapy supports the interest of the work. Together with the correlation with patient data and in vivo (xenografts and PDX) experiments the authors proposed that targeting P4HA1- HIF-1 α axis could contribute to improve TNBC treatment.

We find the manuscript suitable for publication in Nature Communications after addressing the following comments.

Major comments:

- How is P4HA1 induced in TNBC cells? Is it HIF1 α dependent? Are TNBCs more HIF1 α dependent than other breast cancer cell types?
- The results in figure 1a,b suggest that Her2+ tumors express similar levels of P4HA1 than TNBC tumors. Could the authors comment on this?
- Since it is key for the mechanism proposed that PHD levels are not directly affected by the silencing of P4HA1, and even though this is shown in the figure S3 for PHD1 (known not to be upregulated by HIF1), the reviewer would like to see the levels of the three PHDs upon P4HA1 silencing (as PHD2 and PHD3 are HIF targets).
- Could the authors provide the expression level of P4HA1 as a control of the effectiveness of the overexpression and silencing for each concerning western blot?
- Could the authors specify which inhibitor and at which concentration it was used in each figure? Since PHDs could also be inhibited, could the authors comment on the specificity of such inhibitors and how it affects their conclusions? Could the authors explain their choice regarding the inhibitor used in vivo?
- In response to chemotherapy the authors attribute the increase in resistance of the CD44+/CD24-/low (clones 3 and 4) to the increased levels of P4HA1. They proved that silencing P4HA1 rescues the sensitivity to docetaxel and doxorubicin. What is the effect of the silencing in the sensitive clones (clones 1 and 3)? Are they further sensitized? Why do the authors choose the HMLE (breast epithelial) cell line for these experiments instead of the TNBC cell lines (MDA MB231 or Hs578T)? Could they explain in detail how do they obtain the CD44+/CD24-/low and the CD44low/CD24+ clones?
- Interestingly upon short term treatment with docetaxel or doxorubicin the levels of P4HA1 are up-regulated (figure S5). How do the authors explain it? Is it a selection (increased proliferation or decreased apoptosis) of the high expressing cells? Or is it because of a direct effect on the expression of P4HA1 in response to the drugs? What is the time line of these experiments?
- In the in vivo experiments (figure 7 a and b) there are differences on the effect in the primary tumor growth when looking at the effect of the KD vs the P4H inhibitor. How do the authors explain no effects upon silencing but significant reduction in tumor growth in response to the single treatment with the P4H inhibitor?

- The combo (KD+docetaxel) improves the effect of the single treatments but eventually the tumors grow exponentially. Moreover, it seems that the combination of P4Hi+docetaxel is not so effective as the combo KD+docetaxel. Could the authors discuss this and the relevance of the eventual tumor escape?

Minor comments:

- General: minor typing mistakes.
- Figure 1
 - o 1a: could the authors clarify the hormone expression status of Her2 tumors and provide representative images?
 - o 1b: Could the authors clarify the quantification strategy of P4HA1 stainings (scoring method).
 - o 1d: the data on collagen secretion is missing. Could the authors include it on the figure, as stated in the text?
- Figure 2: could the authors specify the O2 levels throughout the figure?
- Figure 3:
 - o 3c: Could the authors provide the loading in inputs controls for the IP?
 - o 3d: could the authors specify in the figure legend/material and methods the duration of bortezomib treatment?
- Figure 4: Could the authors further comment on the significance of the results obtained in figure 4b, and compare it with the information obtained in 4c?
- Figure 5:
 - o 5b: Could the authors provide the protein expression levels of HIF1a?
 - o 5c: could the authors specify how did they normalize?
 - o Is there a difference in % of CD44+/CD24-/low and CD44low/CD24+ in control vs shP4HA1 cells?
 - o 5h,k: unstained controls should be provided. An indication of the gate on ALDH or DCF high cells should be provided. Could the authors provide which instrument have they used for the flow cytometric analysis?
- Figure 6:
 - o 6c: The authors should clarify the normalization strategy used in 6c
 - o How do the authors explain the increase in tumor sphere formation upon Dox or Doc treatment in shControl MDA-MB-231 cells?
- Figure 7:
 - o 7a,b: The reviewer would like to know whether at the start of each treatment animals were randomized in groups with the same average. From the text and the figures, it seems that it is not the case. This is especially relevant in figure 7b.
 - o 7b: arrow indicating the beginning of P4Hi treatment should be provided, since in the M&M is not clear whether the treatment starts right after tumor injection and for 2 weeks or after two weeks, for the rest of the experiment.
 - o In 7b n=8, whereas in the IHC analysis + met analysis n is lower, suggesting that the analysis hasn't been performed in all tumors. The authors should clarify this aspect.
 - o 7c: could the authors provide the quantification of HIF-1a and collagen?
- Figure 8:
 - o 8d: it should be specified which treatment course did the patients follow.
- Supplementary figure 8: if possible, could the authors change the pictures of DHB for ones with the same background/White balance as the rest?

Reviewer #2:

Remarks to the Author:

The manuscript by Xiong et al. shows the role of collagen prolyl hydroxylase P4HA1 in HIF1 α pathway in triple-negative breast cancer (TNBC). The authors propose that P4HA1 controls HIF1 α stability by regulating the intracellular levels of succinate and α -ketoglutarate, which in turn

control HIF-prolyl hydroxylase (PHD) activity. According to their results, this axis P4HA1/HIF1 α plays an important role in the response of TNBC patients to chemotherapy, highlighting the strategy of targeting P4HA1 to improve TNBC treatment.

The story is clearly presented, and robustly sustained by evidence, particularly the *in vivo* experiments. However, the manuscript presents some inconsistencies and technical issues. Specific comments are as follows.

1. All over the manuscript, the authors used inconsistently different cell lines for different groups of experiment. Sometimes, the specific cell line used in the experiment is not even indicated. The authors should be more consistent regarding the cellular model used.
2. In the manuscript, PHDs have been proposed by the authors to play a key role in the connection between P4HA1 and HIF1 α . However, no experiment of lack-of-function or gain-of-function of PHDs has been included in the manuscript. Those experiments will help to elucidate the mechanism of HIF-1 α stabilization by P4HA1.
3. Figure 1D. The authors claimed an increase of collagen secretion but this figure only shows the difference in protein level of P4HA1 between TNBC cell lines and luminal cancer cells.
4. Figure S1. The authors claimed to use shRNA of P4HA1 or P4H inhibitors to check tumor progression in 3D culture systems, but only the result with shRNA against P4HA1 was shown. Data showing pharmacological P4HA1 inhibition (as claimed in the text) should be included in this figure.
5. The authors used gene co-expression and gene ontology analysis but the protocol is missing in the materials and methods part. Please include in methods section the procedure for gene co-expression and gene ontology analysis, as well as details of the statistical analysis of these results.
6. Figure 2B. The authors should include a positive control, namely HIF1 α levels in hypoxia. Moreover, the authors might consider to include collagen secretion upon overexpression of P4HA1 and knockdown of P4HA1.
7. Figure 2D. Why are the levels of HIF1 in these panels so high in comparison to Figure 2E (normoxia)? Have these experiments been performed in normoxic or hypoxic conditions?
8. Figure 2G, 2H and 2I. In experiments showing HRE-luciferase activity and expression of HIF1 α -target genes, positive control of cells incubated in hypoxia should be included.
9. Figure 2F and Figure S2A. The concentration of P4H inhibitors is not consistent.
10. Figure 3E. In comparison with Figure 2D, why a decrease in HIF1 α level was not seen upon shP4HA1 in MDA231 cells? For comparative purposes, the authors should consider to do the overexpression of P4HA1 and shP4HA1 in the same cell line.
11. Figure 3D, 3E, 4F and 4H lack the control immunoblot for P4HA1.
12. Figure 3F and 3G lack a positive control in hypoxia. Again, it would have been convenient to perform these experiments in the same cell line.
13. Figure 3H should include an immunoblot for hydroxylated-HIF1 α .
14. In page 7, the authors claimed that "We have shown that PHD is active in hydroxylating HIF-1 α in these cells when P4HA1 is knocked down, but P4HA1 does not influence the protein level of the PHD (Figure S3)". However, Figure S3 does not show any result regarding PHD activity.
15. Figure S3 only shows the immunoblot against PHD1. Immunoblots against PHD2 and PHD3 should be included as well. Indeed, among all the 3 members of the PHD family, PHD2 is the main isoform regulating HIF1 α stability in cells (although all 3 enzymes can hydroxylate HIF1 α *in vitro*).
16. In page 8, the sentence "We therefore determined whether P4HA1 enhances proline hydroxylation on HIF1 α " would have a more logical meaning if expressed as "We therefore determined whether P4HA1 reduces proline hydroxylation on HIF1 α ".
17. Figure 4F and 4G. The concentration of octyl- α -ketoglutarate and dimethyl-succinate were not indicated.
18. Figure S4A and S4B. The authors should check if octyl- α -ketoglutarate treatment prevents the effect of P4HA1 overexpression (as they did in Figures 4H and 5F).

Remarks to the Author:

Xiong. G et demonstrated the novel role of collagen P4HA1 in promoting chemoresistance by modulating HIF-1 dependent cancer cell stemness in TNBC models, patient derived organoids and patient cohort. Additionally, they demonstrated that the P4HA1 inhibitor or inhibition of P4HA1 in combination with decetaxel treatment induces regression of tumor growth and colonization in TNBC breast cancer in vivo models. This is interesting, new and an important observation. However, authors need to address if high or low expression of P4HA1 correlates to the chemotherapy response in a HIF-1 alpha dependent manner in TNBC patients using publically available multiple datasets to address if P4HA1 is a predictive/prognostic marker in response to TNBC chemotherapy treatment.

Major Specific comments:

1. In fig. 1c, authors need to include P4HA1 staining of metastatic TNBC tumor sections ideally with matched primary tumors to address if P4HA1 expression is associate with TNBC progression. Additionally, they need to ideally compare the P4HA1 expression in metastatic ER+ tumor sections with matched primary tumor sections to address if high P4HA is specific to TNBC progression not ER+ progression. If matched samples cannot be found, then unmatched analysis would be acceptable.
2. In all P4HA1 knockdown experiments performed in TNBC cell lines, lack rescue controls. In all these experiments they need to reconstitute P4HA1 in TNBC knockdown cells and show the effect or phenotype is rescued and is specific to P4HA1. (Fig, 2d, e; 3d, e; 4b-e, g; 5e, g-k; Fig 6a, b, d, f, g and 7a-c).
3. The role of P4HA1 in breast cancer metastasis is well established in the literature so the key contribution here seems to be linking P4HA1 to HIF1 alpha for inducing chemotherapy resistance. They need to address if HIF1 alpha is critical for P4HA1 mediated chemotherapy resistance, cell stemness, and tumor growth using TNBC models by performing HIF1 alpha knockdown experiments.
4. Fig 3a-c, they need to perform similar experiments in appropriate TNBC cell lines instead of 293T cells.
5. They need to show stem cell markers enrichment (CD44/CD24^{low}) in in vivo tumors showed in fig 7a pre and post treatment with decetaxel similar to shown in HMLE clones in fig 6c.
7. In result section: "P4HA1 expression is up-regulated in TNBC and is associated with collagen production", expression comparison of P4HA1 in TCGA/Curtis - TNBC Vs Other subtypes + across TNBC stages is missing. Showing levels of P4HA1 in actual patient data will provide a more realistic perspective of high levels both across and within subtypes.
8. Similarly, lack of association between levels of P4HA1 and HIF-1 α at mRNA level should be tested in patient samples as well.
9. The study shows survival curves from kmplot, which consolidates different datasets together. Performing survival analysis on individual dataset like Curtis et al (METABRIC) by using the therapy annotations for the categorization needs to be done as a confirmation of the finding.
10. Studies in the past have shown importance of STAT3 along with HIF-1 α in chemoresistance in TNBC. The status of STAT3 in P4H41-high and HIF-1 α -stable TNBC cell-lines and samples needs to be examined and discussed.

Minor comments:

1. In Figure 8D and e patient number in respective groups should be shown.
2. Quantification of fig S7 should be done to address the differences between different groups.
3. Fig 1A, ki67 staining should be included in the panel. Additionally, they need to include P4HA1 staining high stage HER2 positive tumor sections with matched primary tumors to address if P4HA1 expression is associate with only TNBC progression not HER2 positive breast cancer progression.
4. Fig. 1d, include Her2 positive breast cancer cell lines (BT474 and SKBR3) and test the endogenous expression of P4HA1.
5. Its not clear why authors used LamA/C as loading controls for whole cell extracts throughout manuscript and all of signals are saturated. Include tubulin or actin or GAPDH and use lighter exposures.
6. Fig. 5c, they also need to show P4HA1 and HFI1 alpha protein expression.
7. Fig 6b, HIF 1 alpha status need to be included.
8. Docetaxel and doxorubicin are the only chemotherapy agents used to study the effect of P4HA1 in TNBC models. They need to include cross linking agent like a platin salt to fully establish resistance phenotype.

I thank reviewers for their very thoughtful and constructive comments, which are most helpful in refining and strengthening this study.

Reviewer #1, Expertise: HIF, metabolism (Remarks to the Author):

In the present study Xiong et al. describe a new mechanism contributing to HIF-1 α stability in TNBC. The authors demonstrate that expression of the collagen prolyl 4-hydroxylase subunit 1 (P4HA1) in cancer cells is essential for HIF-1 α stabilization in an oxygen independent manner. The role of HIF-1 α in breast cancer progression and metastasis has been described before, however the authors shed light on a new mechanism that further contributes to HIF-1 α mediated malignancy in TNBC. Furthermore, an interesting role of the P4HA1-HIF-1 α axis in the resistance to chemotherapy supports the interest of the work. Together with the correlation with patient data and in vivo (xenografts and PDX) experiments the authors proposed that targeting P4HA1- HIF-1 α axis could contribute to improve TNBC treatment.

We find the manuscript suitable for publication in Nature Communications after addressing the following comments.

Response: We thank the reviewer for this and the following comments.

Major comments:

• *How is P4HA1 induced in TNBC cells? Is it HIF1 α dependent? Are TNBCs more HIF1 α dependent than other breast cancer cell types?*

Response: It has been shown that P4HA1 transcription is induced by HIF-1 α and hypoxia in mammary epithelial cells (7, 61). Our data and previous report suggest a positive feedback loop between collagen hydroxylation and the HIF-1 pathway. Our data also suggest that P4HA1 in TNBC is not only regulated by HIF-1 α . We have unpublished data showing that P4HA1 expression was induced during the epithelial-mesenchymal transition (EMT). EMT inducer Twist, Snail, and TGF- β can stimulate P4HA1 expression in mammary epithelial cells. It has been reported that TNBC/basal-like breast cancer is associated with activation of the EMT program. Thus, P4HA1 expression in TNBC may be induced by EMT activators. This discussion has been included in the manuscript on page 17.

[Redacted]

HIF-1 activation is associated with the development and progression of ER positive and HER2 positive breast cancer. The activation also contributes to ER positive and HER2 cancer progression. Gene expression analysis showed that the HIF-1 pathway is hyperactivated in basal-like breast cancer (20). It is not clear if TNBC are more HIF-1 dependent than other breast cancer subtypes. We respectfully suggest this question to be addressed in future.

• *The results in figure 1a, b suggest that Her2+ tumors express similar levels of P4HA1 than TNBC tumors. Could the authors comment on this?*

Response: Thank the reviewer for bringing this to our attention. We also noticed that P4HA1 expression was upregulated in HER2+ tumor. In addition, it has been shown that ERBB2 overexpression in mammary epithelial is associated with P4HA2 expression (Mackay A et al, Oncogene, 2003, 1; 22 (17)). Gilkes DM, *et al.*'s study showed that P4HA1 mRNA levels were increased in HER2 breast cancer (7). We have analyzed P4HA1 mRNA expression in published microarray dataset (supplemental Figure S9a) and confirmed their findings. We now include the following sentences in the abstract “we showed that P4H alpha 1 subunit (P4HA1) protein expression was induced in triple-negative breast cancer (TNBC) and HER2 positive breast cancer” and in the discussion on page 14: ‘P4HA1 expression is also upregulated in HER2 positive tumor. However, roles of P4HA1 in HER2 cancer progression still needs further clarification.’

• *Since it is key for the mechanism proposed that PHD levels are not directly affected by the silencing of P4HA1, and even though this is shown in the figure S3 for PHD1 (known not to be upregulated by HIF1), the reviewer would like to see the levels of the three PHDs upon P4HA1 silencing (as PHD2 and PHD3 are HIF targets).*

Response: This point is well taken. We have examined PHD2 and PHD3 protein levels in control and P4HA1-silenced MDA-MB-231 cells (supplemental Figure S3). We found that PHD3 was expressed at low level in MDA-MB-231 cells. Silencing P4HA1 did not significantly change PHD1 and PHD3 protein expression (as showed in figure S3). PHD2 protein levels are slightly down-regulated in P4HA1-silenced MDA-MB-231 cell.

• *Could the authors provide the expression level of P4HA1 as a control of the effectiveness of the overexpression and silencing for each concerning western blot?*

Response: As the reviewer suggested, we now include the P4HA1 western blot data to confirm the overexpression and silencing efficiency (See revised Figure 2, 3, 4).

• *Could the authors specify which inhibitor and at which concentration it was used in each figure? Since PHDs could also be inhibited, could the authors comment on the specificity of such inhibitors and how it affects their conclusions? Could the authors explain their choice regarding the inhibitor used in vivo?*

Response: We have specified which inhibitor was used in each figure and also provided the concentration information in figure legends. It has been shown that 1,4-DPCA inhibits P4HA1 activity with an IC50 value of 3.6 μ M (Franklin TJ *et al*, Biochem J. 2001 15; 353:333-8). 1,4-DPCA have also been found to inhibit the asparaginyl-hydroxylase, factor inhibiting HIF (FIH), with an IC50 value of 60 μ M (Banerji B et al, Chem Commun (Camb). 2005 Nov 21;(43):5438-40. Epub 2005 Oct 4.). 3,4-dihydroxybenzoic acid (DHB) inhibits P4AH1 activity with an IC50

value of 5 μM , whereas DHB inhibits PHDs at relatively high concentration (250 μM to 500 μM) *in vitro* (Hirsilä M et al, J Biol Chem. 2003 15; 278(33): 30772-80. Characterization of the human prolyl 4-hydroxylases that modify the hypoxia-inducible factor.). In the tissue culture experiment, we showed that 1,4-DPCA (5 μM) or DHB (10 μM) treatment or reduced HIF-1 α protein level and HRE luciferase activity. It is not likely that PHDs activity is inhibited by those inhibitors at these concentrations.

DHB has been safely used *in vivo* to inhibit collagen deposition at 40 mg/kg weight per day (7). It also inhibits cancer metastasis at this concentration (7). Thus, we chose this inhibitor for the xenograft experiments. This rationale has been included in the supplemental material and methods on page 22.

• *In response to chemotherapy the authors attribute the increase in resistance of the CD44+/CD24-/low (clones 3 and 4) to the increased levels of P4HA1. They proved that silencing P4HA1 rescues the sensitivity to docetaxel and doxorubicin. What is the effect of the silencing in the sensitive clones (clones 1 and 2)? Are they further sensitized? Why do the authors choose the HMLE (breast epithelial) cell line for these experiments instead of the TNBC cell lines (MDA MB231 or Hs578T)? Could they explain in detail how do they obtain the CD44+/CD24-/low and the CD44low/CD24+ clones?*

Response: P4HA1 protein levels are relatively low in clones 1 and 2 and already sensitive to docetaxel and doxorubicin; therefore we only performed P4HA silencing and P4H inhibitor treatment experiments in clone 3 and 4 that exhibit high P4HA1 expression. To further confirm that increased P4HA1 expression contributes to the chemoresistance of breast cancer cells, we have done the combined drug treatment experiments in MDA-MB-231 cells (See Figure 6f). P4H inhibitor (1,4-DPCA) treatment also sensitized MDA-MB-231 cells to doxorubicin. These results are consistent with our finding in HMLE cells.

All the CD44+/CD24-/low and the CD44low/CD24+ clones were derived from HMLE cells. In brief, we passaged about 1000 HMLE cells to a 100 mm culture dish. After the mono-cell clones growth, we picked up the clones and expanded them. The CD44/CD24 levels in these clones were examined by FACS analysis. Similar method has also been used by (Mani SA *et al*, Cell. 2008 16;133(4):704-15). We now include this information in supplemental experimental procedure (page 21) and the reference in the manuscript.

• *Interestingly upon short term treatment with docetaxel or doxorubicin the levels of P4HA1 are up-regulated (figure S5). How do the authors explain it? Is it a selection (increased proliferation or decreased apoptosis) of the high expressing cells? Or is it because of a direct effect on the expression of P4HA1 in response to the drugs? What is the time line of these experiments?*

Response: We treated the TNBC cells with relatively low concentration of docetaxel (0.1 nM) and doxorubicin (10 nM) for about 4-7 days. It has been reported that cancer stem cells or tumor-initiating cells are more tolerant to docetaxel and doxorubicin treatment (26, 37) Thus, docetaxel and doxorubicin treatment may select cancer cells with high stemness from TNBC cell lines. We found that P4HA1 expression is increased in the cancer stem cell-enriched cell population

(Figure 5a, b, c). Therefore, short term treatment with docetaxel or doxorubicin may enrich cancer stem cells with high levels of P4HA1. We have included this discussion in the manuscript on page 11.

• *In the in vivo experiments (figure 7 a and b) there are differences on the effect in the primary tumor growth when looking at the effect of the KD vs the P4H inhibitor. How do the authors explain no effects upon silencing but significant reduction in tumor growth in response to the single treatment with the P4H inhibitor?*

Response: Thank the reviewer for bringing this to our attention. We also noticed that the P4H inhibitor is more potent to inhibit primary tumor growth. P4HA family includes P4HA1, P4HA2 and P4HA3. Inhibitory activity of DHB is not specific to P4HA1 (Vasta JD, ACS Chem Biol. 2016 15; 11(1):193-9). It is possible that P4H inhibitor treatment suppresses tumor growth by inhibiting other P4HA family members. We now include this discussion on page 12.

• *The combo (KD+docetaxel) improves the effect of the single treatments but eventually the tumors grow exponentially. Moreover, it seems that the combination of P4Hi+docetaxel is not so effective as the combo KD+docetaxel. Could the authors discuss this and the relevance of the eventual tumor escape?*

Response: The reviewer is right that the silencing P4HA1 was more effective to sensitize MDA-MB-231 cells to docetaxel treatment compared with P4Hi. One difference between shRNA experiment and DHB treatment is that P4HA1 was consistently knockdown by shRNA, while DHB treatment was stopped at day20. It is plausible that continually silencing P4HA1 delayed the relapse after the docetaxel treatment. Another possibility is that silencing P4HA1 may be more potent in reducing P4HA1 activity. Therefore, it is crucial to identify more potent and specific P4HA1 inhibitors in the future.

Minor comments:

• *General: minor typing mistakes.*

Response: We apologize for these oversights. The typing mistakes have been corrected.

• *Figure 1*

o 1a: could the authors clarify the hormone expression status of Her2 tumors and provide representative images?

Response: We now include images of HER2 positive tumor samples and status of hormone expression in the revised Figure 1A.

o 1b: Could the authors clarify the quantification strategy of P4HA1 staining (scoring method).

Response: Protein levels of P4HA1 expression in human breast cancer tissues were determined using tissue microarray (TMA). Each case in the TMA is represented by three separate tissue cores. IHC scoring results from each of these three tissue cores were averaged to produce a final

score for each case. P4HA1 immunohistochemical staining was scored according to a standard semi-quantitative scale as follows: negative (0), weak (1), moderate (2), strong (3). Scoring was performed by a pathologist who was blinded to clinical and pathologic variables. This information is now included in the Supplemental Material and Methods on page 18.

o 1d: the data on collagen secretion is missing. Could the authors include it on the figure, as stated in the text?

Response: The collagen secretion data have been included in the revised Figure 1d.

• *Figure 2: could the authors specify the O₂ levels throughout the figure?*

Response: We have specified the O₂ levels 2 in figure2.

• *Figure 3:*

o 3c: Could the authors provide the loading in inputs controls for the IP?

Response: As reviewer suggested, we have provided the loading controls (β -Tubulin) for the input (See Figure 3c).

o 3d: could the authors specify in the figure legend/material and methods the duration of bortezomib treatment?

Response: We have specified both in the figure legend that the duration of bortezomib treatment is 12 hours.

• *Figure 4: Could the authors further comment on the significance of the results obtained in figure 4b, and compare it with the information obtained in 4c?*

Response: The SIRM experiment has been used to identify metabolite changes at the cellular level as an unbiased approach, and *n* value is not big enough for the statistic significant analysis. We hypothesize that collagen hydroxylation mainly consumes the cytoplasmic α -KG, thus we specifically measured cytoplasmic α -KG level in 4c.

• *Figure 5:*

o 5b: Could the authors provide the protein expression levels of HIF1 α ?

Response: We have analyzed the HIF-1 α protein levels in those clones and data are now included in Figure 5b. We found that HIF-1 α protein levels were up-regulated in clone3 and 4 compared to clone1 and 2.

o 5c: could the authors specify how did they normalize?

Response: The gene expression values were derived from the published microarray dataset. The data were log₂ transformed and mean centered. This information has been included in the figure legend, and the reference has been cited in the result section on page 9.

o Is there a difference in % of CD44+/CD24-/low and CD44low/CD24+ in control vs shP4HA1 cells?

Response: We have analyzed cell surface levels of CD44 and CD24 using FACS. Results showed that the CD44+/CD24^{-/low} cell population is significantly lower in shP4HA1 MDA-MB-231 cells compared with control cells (as showed in Figure 5d).

o 5h,k: unstained controls should be provided. An indication of the gate on ALDH or DCF high cells should be provided. Could the authors provide which instrument have they used for the flow cytometric analysis?

Response: The unstained controls and the gate indications have been included in the revised Figure 5h,k. The instrument information has been included in supplemental material and methods.

• *Figure 6:*

o 6c: The authors should clarify the normalization strategy used in 6c

Response: Cell number in control, docetaxel and doxorubicin treated groups were analyzed with the Cell Counting Kit-8 kit (CCK-8, Sigma Aldrich, 96992). For each cell clone, the OD450 values in the drug treatment group were normalized to the value in the control group. We now include this information in the supplemental material and methods on page 21 and 22.

o How do the authors explain the increase in tumor sphere formation upon Dox or Doc treatment in shControl MDA-MB-231 cells?

Response: It has been reported that tumor-initiating cells are more resistant to the chemotherapeutic agent treatment (26, 37). It is most likely that Dox or Doc treatment selected the tumor-initiating cell population in shControl MDA-MB-231 cells, and consequently enhances tumor sphere formation. We now include this discussion on page 11.

• *Figure 7:*

o 7a,b: The reviewer would like to know whether at the start of each treatment animals were randomized in groups with the same average. From the text and the figures, it seems that it is not the case. This is especially relevant in figure 7b.

Response: Mice were randomly grouped before the treatment for all of the *in vivo* experiments, and tumor sizes of each group were similar at Day 1 in the figure. As for Figure 7b, mice were

treated with P4Hi once a day from Day 1 and the treatment may slow down tumor growth. The tumor volume in Doc treatment group was slightly smaller than the control group on Day 5, but no significant difference was detected.

o 7b: arrow indicating the beginning of P4Hi treatment should be provided, since in the M&M is not clear whether the treatment starts right after tumor injection and for 2 weeks or after two weeks, for the rest of the experiment.

Response: We now include the detail information for our animal experiment in the figure legend, Materials/Methods on page 18, and supplemental information on page 22. We measured tumor size at about 2 weeks after tumor injection, which was marked as Day 1 in the tumor growth curve. Since Day 1, P4Hi group mice were treatment with P4Hi once a day, during the whole Doc treatment period (from Day 1 to Day 20).

o In 7b n=8, whereas in the IHC analysis + met analysis n is lower, suggesting that the analysis hasn't been performed in all tumors. The authors should clarify this aspect.

Response: We are sorry that we did not clarify it clearly in the manuscript. These are two experiments. For the experiment in 7b, the tumors were kept in mice after 21 days to examine the tumor regrowth without drug treatment. For the experiment in 7c-f, to examine the effects of drug treatments on cell proliferation and apoptosis, we performed surgery to remove tumors at Day 21 for H&E and IHC analyses. At this time point, tumor did not start to regrown and the drug may still be effective. The mice were kept for another month after the surgery to detect metastasis to lung. We now add this information in the Supplemental Material and Methods on page 22.

o 7c: could the authors provide the quantification of HIF-1 α and collagen?

Response: This point is well-taken. We have quantified the HIF1 α and collagen in Figure 7c. The quantification results are showed in supplemental Figure S7c, d.

• *Figure 8:*

o 8d: it should be specified which treatment course did the patients follow.

Response: The data were derived from the published microarray datasets (48). It includes both adjuvant and neoadjuvant chemotherapy. This information is now included in the figure legend.

• *Supplementary figure 8: if possible, could the authors change the pictures of DHB for ones with the same background/White balance as the rest?*

Response: As the reviewer suggested, we have replaced the IHC pictures of DHB group with similar background/White balance as the rest.

Reviewer #2, Expertise: Prolyl hydroxylase, metabolism (Remarks to the Author):

The manuscript by Xiong et al. shows the role of collagen prolyl hydroxylase P4HA1 in HIF1 α pathway in triple-negative breast cancer (TNBC). The authors propose that P4HA1 controls HIF1 α stability by regulating the intracellular levels of succinate and α -ketoglutarate, which in turn control HIF-prolyl hydroxylase (PHD) activity. According to their results, this axis P4HA1/HIF1 α plays an important role in the response of TNBC patients to chemotherapy, highlighting the strategy of targeting P4HA1 to improve TNBC treatment.

The story is clearly presented, and robustly sustained by evidence, particularly the in vivo experiments. However, the manuscript presents some inconsistencies and technical issues. Specific comments are as follows.

Response: We thank the reviewer for finding the study is clearly presented and robustly sustained by evidence.

1. All over the manuscript, the authors used inconsistently different cell lines for different groups of experiment. Sometimes, the specific cell line used in the experiment is not even indicated. The authors should be more consistent regarding the cellular model used.

Response: We apologize for not indicating the cell lines in some experiment. We found that P4HA1 is highly expressed in TNBC cell lines MDA-MB-231, MDA-MB-157, and Hs579T compared with MCF10A and HMLE (see response to comment 9). For the loss of function experiments, we silenced P4HA1 in MDA-MB-231 and Hs578T cells. For the gain of function experiment, we introduced P4HA1/P4HB in non-malignant MCF10A and HMLE cell lines. To address reviewer's concerns about cell consistency, we have performed the *in vitro* drug sensitivity experiments and the HIF-1 α stability experiments in MDA-MB-231 cells. The data are now included in Figure 3a, b and Figure 6f.

2. In the manuscript, PHDs have been proposed by the authors to play a key role in the connection between P4HA1 and HIF1 α . However, no experiment of lack-of-function or gain-of-function of PHDs has been included in the manuscript. Those experiments will help to elucidate the mechanism of HIF-1 α stabilization by P4HA1.

Response: This point is well-taken. We first examined whether PHDs protein levels are modulated by P4HA1. We found that knockdown of P4HA1 had little effect on PHD1 and 3 protein expression. PHD2 expression was slightly reduced in P4HA1-silenced cells. Since PHD1 and PHD2 are two major isoforms expressed in MDA-MB-231 cells (Figure s3a), we have performed the lack-of-function experiments in P4HA1-silenced cells to examine if silencing PHD1 or PHD2 can restore HIF-1 α protein expression. As shown in supplemental Figure S3c, silencing PHD2 restored HIF-1 α protein levels in P4HA1-silenced MDA-MB-231 cells. These experiments further confirm that P4HA1 increases HIF-1 α protein level by reducing PHD2-dependent proline hydroxylation.

3. *Figure 1D. The authors claimed an increase of collagen secretion but this figure only shows the difference in protein level of P4HA1 between TNBC cell lines and luminal cancer cells.*

Response: We are sorry for this oversight. The collagen secretion data are now included in Figure 1d.

4. *Figure S1. The authors claimed to use shRNA of P4HA1 or P4H inhibitors to check tumor progression in 3D culture systems, but only the result with shRNA against P4HA1 was shown. Data showing pharmacological P4HA1 inhibition (as claimed in the text) should be included in this figure.*

Response: We have included these data in Figure S1d-f.

5. *The authors used gene co-expression and gene ontology analysis but the protocol is missing in the materials and methods part. Please include in methods section the procedure for gene co-expression and gene ontology analysis, as well as details of the statistical analysis of these results.*

Response: The protocol of gene co-expression and gene ontology analysis is now included in the supplemental table 1 on page 16.

6. *Figure 2B. The authors should include a positive control, namely HIF1 α levels in hypoxia. Moreover, the authors might consider to include collagen secretion upon overexpression of P4HA1 and knockdown of P4HA1. (Not necessary)*

Response: We have examined HIF-1 α protein level in control and P4H1 overexpression cells under hypoxia condition, and the data are now include in Figure 2b. Collagen secretion in P4H1 overexpression cells was upregulated and downregulated in P4HA1 silencing cells as shown in Figure S2c.

7. *Figure 2D. Why are the levels of HIF1 in these panels so high in comparison to Figure 2E (normoxia)? Have these experiments been performed in normoxic or hypoxic conditions?*

Response: These blots were from two different experiments. The HIF-1 α protein level was very high in hypoxia condition, and thus the bot in Figure 2f (Original 2e) was exposed in much short time compared with the blot in Figure 2e (original 2d). Therefore, we cannot compare these two blots because they were generated under different exposure conditions.

8. *Figure 2G, 2H and 2I. In experiments showing HRE-luciferase activity and expression of HIF1 α -target genes, positive control of cells incubated in hypoxia should be included.*

Response: This point is well taken. We have performed the HRE-luciferase report assay and examined HIF1 α -target genes expression under hypoxia condition. The results are now included in Figure 2h, i, j.

9. Figure 2F and Figure S2A. The concentration of P4H inhibitors is not consistent.

Response: We apologize for the wrong labeling in Figure S2d. The concentration is at μ M.

10. Figure 3E. In comparison with Figure 2D, why a decrease in HIF1 α level was not seen upon shP4HA1 in MDA231 cells? For comparative purposes, the authors should consider to do the overexpression of P4HA1 and shP4HA1 in the same cell line.

Response: We tried to compare HIF-1 hydroxylation at similar amount of total HIF-1 α protein, thus we loaded more protein lysate for shP4HA1 MDA-MB-231 cells, as shown by the loading control Lamin A/C.

We now provide a rationale for performing gene silencing and overexpression in different cell lines. See the response to comment 1. In general, we silenced P4HA1 in cell lines with high P4HA1 expression level (MDA-MB-231), and ectopically expressing P4HA1 in cells lines with low P4HA1 protein levels (MCF10A and HMLE).

11. Figure 3D, 3E, 4F and 4H lack the control immunoblot for P4HA1.

Response: As reviewer requested, we now include the immunoblot for P4HA1 in Figure 3e, 3f (original 3d), 3g (original 3e), 3j, 4f, 4g, and 4j.

12. Figure 3F and 3G lack a positive control in hypoxia. Again, it would have been convenient to perform these experiments in the same cell line.

Response: We have performed these experiments under both normoxia and hypoxia conditions, and the results are now included in Figure 3h (original 3f) and 3i (original 3g).

13. Figure 3H should include an immunoblot for hydroxylated-HIF1 α .

Response: Proline hydroxylation on HIF-1 α P402 and P564 are well characterized (Jaakkola P *et al*, Science 292, 468–472 *et al*; Ivan M, Science 292, 464–468; Masson N *et al*, EMBO J. 20, 5197–5206), and it has been shown that HIF-1 α PPAA mutant is more resistant to hydroxylation-dependent protein degradation (13). We respectfully suggest not repeating these experiments.

14. In page 7, the authors claimed that “We have shown that PHD is active in hydroxylating HIF-1 α in these cells when P4HA1 is knocked down, but P4HA1 does not influence the protein level of the PHD (Figure S3)”. However, Figure S3 does not show any result regarding PHD activity.

Response: We are sorry for the inaccurate description. We now change the sentence to “We showed that silencing P4HA1 had little effect on the protein level of the PHD1 and PHD3, and slightly reduced PHD2 protein level (Figure S3a), suggesting that reduced HIF-1 hydroxylation is not due to increased PHDs expression.” (See page 7).

15. Figure S3 only shows the immunoblot against PHD1. Immunoblots against PHD2 and PHD3 should be included as well. Indeed, among all the 3 members of the PHD family, PHD2 is the main isoform regulating HIF1 α stability in cells (although all 3 enzymes can hydroxylate HIF1 α *in vitro*).

Response: We appreciate that this part of the study need more attention. We have examined all three PHDs protein levels in P4HA silencing MDA-MB-231 cell (Supplemental Figure S3a).

16. In page 8, the sentence “We therefore determined whether P4HA1 enhances proline hydroxylation on HIF1 α ” would have a more logical meaning if expressed as “We therefore determined whether P4HA1 reduces proline hydroxylation on HIF1 α ”.

Response: Thank the reviewer for identifying this mistake. We have corrected the sentence in the revised manuscript.

17. Figure 4F and 4G. The concentration of octyl- α -ketoglutarate and dimethyl-succinate were not indicated.

Response: We now indicate concentration of octyl- α -ketoglutarate and dimethyl-succinate in Figure 4f and 4g.

18. Figure S4A and S4B. The authors should check if octyl- α -ketoglutarate treatment prevents the effect of P4HA1 overexpression (as they did in Figures 4H and 5F).

Response: We have performed the seahorse assay in P4HA1-expressing MCF10A cells in the presence of octyl- α -ketoglutarate (See FigureS5b), and results showed that octyl- α -ketoglutarate treatment rescued oxygen consumption in P4HA1 overexpression cells.

Reviewer #3, Expertise: Breast cancer, drug resistance (Remarks to the Author):

Xiong. G et demonstrated the novel role of collagen P4HA1 in promoting chemoresistance by modulating HIF-1 dependent cancer cell stemness in TNBC models, patient derived organoids and patient cohort. Additionally, they demonstrated that the P4HA1 inhibitor or inhibition of P4HA1 in combination with decetaxel treatment induces regression of tumor growth and colonization in TNBC breast cancer in vivo models. This is interesting, new and an important observation. However, authors need to address if high or low expression of P4HA1 correlates to the chemotherapy response in a HIF-1 alpha dependent manner in TNBC patients using publically available multiple datasets to address if P4HA1 is a predictive/prognostic marker in response to TNBC chemotherapy treatment.

Response: We appreciate the reviewer finds the study interesting, new, and important.

Major Specific comments:

1. In fig. 1c, authors need to include P4HA1 staining of metastatic TNBC tumor sections ideally with matched primary tumors to address if P4HA1 expression is associate with TNBC progression. Additionally, they need to ideally compare the P4HA1 expression in metastatic ER+ tumor sections with matched primary tumor sections to address if high P4HA is specific to TNBC progression not ER+ progression. If matched samples cannot be found, then unmatched analysis would be acceptable.

Response: To address this comment, we have analyzed the published microarray dataset generated from 20 primary breast cancer tissues and corresponding metastatic lymph node tumors (Suzuki M, Mol Oncol. 2007 1(2):172-80.). We did not detect a significant difference in P4HA1 expression between the primary tumors and metastasis lesions; in fact, P4HA1 expression is slightly downregulated in the lymph node tumors. It has been reported that P4HA1 expression is essential for breast cancer metastasis (7). These results suggest that the gene promoting cancer metastasis is not necessarily highly expressed at the metastasis lesions. This is especially the case for the EMT-related gene. It is well-established that EMT promotes cancer metastasis by enhancing cancer cell invasion. However, the mesenchymal to epithelial transition (MET) is required for cancer cells to regrow and form macrometastasis (Tsai JH *et al*, Cancer Cell. 2012 Dec 11;22(6):725-36), thus the EMT-related gene and phenotypes may be downregulated in the metastatic lesion (Yao D *et al*. Mol Cancer Res. 2011 9(12):1608-20; Chao YL *et al*, Mol Cancer. 2010 Jul 7;9:179. doi: 10.1186/1476-4598-9-179.). We have some unpublished data showing that P4HA1 expression is induced by Twist during EMT (see response to reviewer 1's comment). It is mostly likely that increased P4HA1 expression in primary tumor promotes cancer cell invasion (7) and stemness (Figure. 5), which facilitates the initiation of cancer cell colonization at the distance organs. However, P4HA1 expression may be repressed during the mesenchymal-epithelial transition (MET) process that is required for the macrometastasis formation. It would be interesting to compare the P4HA1 protein levels

between the matched primary tumors and metastatic lesions in specific subtypes; unfortunately, we don't have the matched samples collected in our institute to address this question.

2. In all *P4HA1* knockdown experiments performed in TNBC cell lines, lack rescue controls. In all these experiments they need to reconstitute *P4HA1* in TNBC knockdown cells and show the effect or phenotype is rescued and is specific to *P4HA1*. (Fig, 2d, e; 3d, e; 4b-e, g; 5e, g-k; Fig 6a, b, d, f, g and 7a-c).

Response: Thank the reviewer for the suggestion. We have reconstituted *P4HA1* expression in sh*P4HA1*-expressing MDA-MB-231 cells (see supplemental information on page 17). We found that restoring *P4HA1* expression rescued HIF-1 α protein level in sh*P4HA1*-expressing MDA-MB-231 cells (Figure S2b). Increased α -KG level and reduced succinate levels in sh*P4HA1*-expressing cells were also rescued by *P4HA1* restoration (Figure S4a, b). We also analyzed cancer cell stemness (tumorsphere formation assay, FACS analysis of aldehyde dehydrogenase activity) and ROS level after *P4HA1* expression was restored in sh*P4HA1* expressing cells (Figure S5a, b, e). We found that cancer cell stemness and ROS levels were rescued by *P4HA1* expression. These results indicate that the effects or phenotypes in sh*P4HA1*-expressing cells are specific to *P4HA1* silence.

3. The role of *P4HA1* in breast cancer metastasis is well established in the literature so the key contribution here seems to be linking *P4HA1* to HIF1 alpha for inducing chemotherapy resistance. They need to address if HIF1 alpha is critical for *P4HA1* mediated chemotherapy resistance, cell stemness, and tumor growth using TNBC models by performing HIF1 alpha knockdown experiments.

Response: As the reviewer pointed out, *P4HA1* roles in cancer metastasis is well established; thus the present study focus on function of *P4HA1* in regulating the HIF-1 pathway and chemoresistance. To address if HIF-1 α is critical for *P4HA1* mediated chemotherapy resistance and cell stemness, we have performed a number of rescue experiments by restoring HIF-1 α protein expression in *P4HA1*-silenced cells. We found that expression of the stable HIF-1 α (PPAA) could at least partially rescue cancer cell stemness and reduced ROS levels and oxidative phosphorylation in *P4HA1*-silenced cells (See Figure. 5h, 5i, 5j, 5k, 5l). To determine whether *P4HA1* promotes chemoresistance through HIF-1, we have knocked down *P4HA1* in HIF-1 α (PPAA)-expressing MDA-MB-231 cells, and then examined the response of those cells

to docetaxel treatment *in vivo*. We found that silence of P4HA1 fails to sensitize PPAA-expressing cells to docetaxel (See Figure S7b). These results indicate that HIF-1 α is critical for P4HA1-mediated cell stemness and chemotherapy resistance. It has been reported that inhibition of the HIF-1 pathway sensitizes TNBC cells to chemotherapy resistance in TNBC and inhibits cancer cell stemness (Zhang C et al, Proc Natl Acad Sci U S A. 2016 5;113(14):E2047-56.), thus we did not perform HIF-1 α knockdown experiments to repeat previous studies. We hope that the rescue experiments in the revised manuscript can address the reviewer's concern.

4. *Fig 3a-c, they need to perform similar experiments in appropriate TNBC cell lines instead of 293T cells.*

Response: We agree. We have performed the HIF-1 α protein degradation experiments in MDA-MB-231 cells, and the results also show that knockdown of P4HA1 significantly increases the HIF-1 degradation at the early time points (Figure 3a). To analyze HIF-1 α ubiquitination, we need to co-transfect three vectors expressing shP4HA1, HIF-1 α -flag, and Ubiquitin-HA into cells (Figure 3e). We encountered a technical issue when performing the HIF-1 α ubiquitination experiments in MDA-MB-231 cells. The transfection efficiency is relatively low in MDA-MB-231 cells compared with 293FT cells, and it is hard to pull down enough protein to analyze HIF-1 α ubiquitination.

5. *They need to show stem cell markers enrichment (CD44/CD24^{low}) in in vivo tumors showed in fig 7a pre and post treatment with docetaxel similar to shown in HMLE clones in fig 6c.*

Response: As the reviewer suggested, we have analyzed expression of the stem cell marker in tumor samples. ALDH has been widely used as a stem cell marker in tumor tissue (Miyata T et al, Anticancer Res. 2017; 37(5):2541-2547. Tomita H et al, Oncotarget. 2016 8;7(10):11018-32.), we now include ALDH IHC staining for the tumor samples in Figure 7a. Results show that silencing P4HA1 reduced ALDH positive staining (supplemental Figure S7a), suggesting that cancer cell stemness is enhanced by P4HA1 *in vivo*.

7. *In result section: "P4HA1 expression is up-regulated in TNBC and is associated with collagen production", expression comparison of P4HA1 in TCGA/Curtis - TNBC Vs Other subtypes + across TNBC stages is missing. Showing levels of P4HA1 in actual patient data will provide a more realistic perspective of high levels both across and within subtypes.*

Response: We appreciate that this part needs more attention. Our human breast cancer TMA data (Figure 1a) indicate that P4HA1 is highly expressed in both TNBC and HER2 breast cancer tissue. Therefore, we have changed the sentence to "The upregulation of P4HA1 in breast cancer cell lines is associated with increased secretion of collagen". As reviewer suggested, we have analyzed the P4HA1 mRNA levels in TCGA dataset. Results showed that P4HA1 is also highly expressed in highly proliferative ER positive breast cancer, and that Her2 positive cancer has highest expression of P4HA1 (Supplemental Figure S9). One potential explanation for the discrepancy between the microarray and IHC data is that we only quantified P4HA1 protein

levels in cancer cell for the IHC analysis, while P4HA1 mRNA expression in TCGA data may derived from both stromal and cancer cells.

8. Similarly, lack of association between levels of P4HA1 and HIF-1 α at mRNA level should be tested in patient samples as well.

Response: As reviewer suggested, we have analyzed the association between P4HA1 and HIF-1 α mRNA levels in the published microarray dataset (Chin K, 2007, Cancer Cell, 10 (6), 529-41). We identified a weak correlation ($r=0.218$) between P4HA1 and HIF-1 α mRNA expression (see below). Our study mainly focuses on the regulation of HIF-1 α at the protein level. Expression of the HIF-1 gene signature reflects HIF-1 activity in cancer tissue. Therefore, we determined the association between P4HA1 expression and HIF-1 gene signature, and results showed that expression of HIF-1 gene signature is associated with P4HA1 levels in human breast cancer tissue (Figure 8f).

9. The study shows survival curves from *kmplot*, which consolidates different datasets together. Performing survival analysis on individual dataset like Curtis et al (METABRIC) by using the therapy annotations for the categorization needs to be done as a confirmation of the finding.

Response: Thank the reviewer for the suggestion and providing the detail information for the analysis. We have performed the survival analysis in the METABRIC dataset, and results confirm the finding that increased P4HA1 expression is associated with poor prognosis after chemotherapy. The data is now included in supplemental Figure S8g.

10. Studies in the past have shown importance of STAT3 along with HIF-1 α in chemoresistance in TNBC. The status of STAT3 in P4HA1-high and HIF-1 α -stable TNBC cell-lines and samples needs to be examined and discussed.

Response: Roles of STAT3 in chemoresistance are well established. Currently we don't have strong data to support that P4HA1 promotes chemoresistance through STAT3. Our unbiased gene co-expression analysis did not detect the association between P4HA1 and the STAT3

pathway or gene signature. In addition, the western blot data show silencing P4HA1 only slightly reduced STAT3 phosphorylation (See below). We now include this discussion on page 16 and 17.

Minor comments:

1. In Figure 8D and e patient number in respective groups should be shown.

Response: The patient number is now included in Figure 8d.

2. Quantification of fig S7 should be done to address the differences between different groups.

Response: As reviewer suggested, we have quantified IHC staining, and data is included in Figure S8f.

3. Fig 1A, ki67 staining should be included in the panel. Additionally, they need to include P4HA1 staining high stage HER2 positive tumor sections with matched primary tumors to address if P4HA1 expression is associate with only TNBC progression not HER2 positive breast cancer progression.

Response: As reviewer requested, we now include Ki67 staining and the Her2 panel in Figure 1A. As we mentioned in the response to comment 1, we don't have the access to the matched HER2 positive breast cancer tissue samples. However, we have analyzed association between P4HA1 expression and prognosis of HER2 breast cancer patients. We found that increased P4HA1 expression correlates with poor prognosis (see below). Therefore, it is most likely that P4HA1 expression is also associated with progression of HER2 positive breast cancer.

4. Fig. 1d, include Her2 positive breast cancer cell lines (BT474 and SKBR3) and test the endogenous expression of P4HA1.

Response: We have compare P4HA1 protein level in Her2 positive breast cancer cell line BT474 with TNBCs MDA-MB-231 cells, MDA-MB-157 cells, BT549 cells and Hs578T cells (Figure S9b). We found that the HER2 positive cancer cell line also have high expression of P4HA1. This result is consistent with the IHC data. However, function of P4AH1 in HER2 positive cancer remains to be determined. We now include this discussion on page 14. See also our response to reviewer 1 comment 2.

5. Its not clear why authors used LamA/C as loading controls for whole cell extracts throughout manuscript and all of signals are saturated. Include tubulin or actin or GAPDH and use lighter exposures.

Response: As reviewer suggest, we have used beta-tubulin as loading controls for new added western blot experiments (Figure 2c, 3a, 3e) and add tubulin as a control for some old experiments (Figure 1d, 2d, 3f, 3g, 4f, 4g, 4j). In addition, we have compared LamA/C and tubulin levels in a number of cell lysates, and found the levels of these two proteins are correlated. In fact, lamin A/C has been used as a loading control in many studies (Jiang S et al, Dis Model Mech. 2015;8(9):1121-7; Scharl M et al, J Biol Chem. 2009 9;284(41):27952-63.), especially for nuclear protein and transcription factor.

6. Fig. 5c, they also need to show P4HA1 and HIF1 alpha protein expression.

Response: The data in Figure 5c were generated from the published expression profile. We are sorry that we cannot analyze the protein expression because we do not have the tissue and cell samples.

7. Fig 6b, HIF 1 alpha status need to be included.

Response: As the reviewer suggested, we have performed HIF-1 α IHC staining in lung metastasis cancer lesions and also found a decreased HIF-1a expression in the P4HA1-silenced group (supplemental Figure S6a).

8. Docetaxel and doxorubicin are the only chemotherapy agents used to study the effect of P4HA1 in TNBC models. They need to include cross linking agent like a platin salt to fully establish resistance phenotype.

Response: To address this comment, we have performed the Cisplatin treatment experiments in MDA-MB-231 cells. Silence P4HA1 or P4H inhibitor (1,4-DPCA) treatment also sensitized MDA-MB-231 cells to Cisplatin (See Figure S6c, d).

Reviewers' Comments:

Reviewer #1:

None

Reviewer #2:

Remarks to the Author:

The authors correctly addressed my comments. I have no further questions.

Reviewer #3:

Remarks to the Author:

Authors have addressed most concerns satisfactorily. Two minor concerns remains:

- 1) In response to the concern that P4HA1 staining in metastatic vs primary tumors should be conducted to test whether P4HA1 levels associate with TNBC progression, authors argue that since P4HA1 is associated with EMT, they do not expect to identify higher P4HA1 levels in metastatic sites. While this may be true, such a claim would be best substantiated by analysis demonstrating that P4HA1 levels in primary tumors associate with a higher likelihood of metastatic progression. Including this analysis will greatly strengthen the clinical relevance of this manuscript.
- 2) In response to the request that the authors show stem cell markers enrichment (CD44/CD24^{low}) in in vivo tumors pre and post treatment with docetaxel, authors present ALDH data. However, the use of ALDH as a stem cell marker has been controversial and CD24/44 levels are considered the gold standard in the field. We recommend the authors assess CD24/44 rather than ALDH levels in this figure.

Decision letter

The present version of the manuscript from Xiong et al. has been notably improved. We appreciate that the authors have tried to answer all our major and minor concerns. However, some of them haven't been fully addressed:

Response: We appreciate that the reviewers found that the manuscript was significantly improved in the last revision. We are grateful to the reviewer for their thoughtful and constructive comments, and below we addressed the reviewers' comments point by point with exact changes in the manuscript.

Comments and response from last submission

• *How is P4HA1 induced in TNBC cells? Is it HIF1 α dependent? Are TNBCs more HIF1 α dependent than other breast cancer cell types?*

Response: It has been shown that P4HA1 transcription is induced by HIF-1 α and hypoxia in mammary epithelial cells (7, 61). Our data and previous report suggest a positive feedback loop between collagen hydroxylation and the HIF-1 pathway. Our data also suggest that P4HA1 in TNBC is not only regulated by HIF-1 α . We have unpublished data showing that P4HA1 expression was induced during the epithelial-mesenchymal transition (EMT). EMT inducer Twist, Snail, and TGF- β can stimulate P4HA1 expression in mammary epithelial cells. It has been reported that TNBC/basal-like breast cancer is associated with activation of the EMT program. Thus, P4HA1 expression in TNBC may be induced by EMT activators. This discussion has been included in the manuscript on page 17.

[Redacted]

Reviewer comments: *even though the authors have tried to link the modulation of P4HA1 expression in TNBC to EMT in a HIF-dependent manner it still remains unclear if this is the case. Thus, the reviewer believes this should be experimentally addressed by performing a HIF-1 α KD experiment in TNBC cells and assessing P4HA1 expression.*

Response: As reviewer suggested, we have performed HIF-1 α knockdown experiments in MDA-MB-231 cells. Our data showed that silencing HIF-1 α reduced P4HA1 protein levels in the TNBC cell line, but P4HA1 expression still remained at the certain level in the HIF-1 α -silenced cells. These results suggest that HIF-1 is not the only pathway that regulates P4HA1 expression during breast cancer progression. In the last rebuttal, we showed the increased P4HA1 expression during EMT. To determine if EMT induces P4HA1 expression in a HIF-dependent manner, we have further silenced HIF-1 α in Twist-induced EMT cells. Consistent with previous report (Yeh YH, Oncol Rep. 2016; 35(5):2887), we found that HIF-1 α levels were upregulated in the EMT cells; however, silence of HIF-1 α in the EMT cells could not completely block Twist-induced P4HA1 expression. By analyzing published ChIP-Seq data (Chang AT, Genes & development;29(6):603), we also found that Twist bound to the regulatory region of P4HA1 gene (fold enrichment 19.26, p-value 1e-92.11). These results suggest that Twist is another transcription factor that directly regulates P4HA1 expression. In the current manuscript, we mainly focus on how P4HA1 regulates the HIF-1 pathway and promotes chemoresistance in TNBC; therefore, we did not include those data in the manuscript. In the future, we will further define the molecular mechanism by which P4HA1 is regulated during EMT and in TNBC.

[Redacted]

Reviewer comments: the authors have explained why they only performed drug sensitivity experiments upon P4HA1 silencing in the clones 3 and 4. However, since the clones 1 and 2 are not included in these experiments (KD and inhibition) It's difficult to see that the initial differences in sensitivity to chemotherapy are due to the P4HA1 expression and not to other factors not analyzed.

Response: This point is well taken. We have performed drug sensitivity experiments in CD44⁻/CD24⁺ clone 1 and clone 2. CD44⁻/CD24⁺ clone 1 and clone 2 are more sensitive to docetaxel and doxorubicin compared with CD44⁺/CD24^{-low} clone 3 and clone 4. Silence of P4HA1 or treatment with P4H inhibitor 1,4-DPCA did not significantly increase their sensitivity to docetaxel and doxorubicin. These results are now included in supplemental Figure 6 (Figure S6c).

Reviewer comments: It still remains unclear the duration of the treatment for the western blot analysis. Is it 4 or 7 days? In any case, in contrast to what is stated in the main text, this duration does not reflect a short-term effect. What is the effect of 24h treatment?

Response: Thank the reviewer for bringing this to our attention. We now clarify in the result and supplemental figure legend that the duration of the treatment is 7 days (on page 12 and Supplemental information page 5), and delete the 'short-term' in the text. We have also examined the effect of 24h docetaxel and doxorubicin treatment on P4HA1 expression. Three TNBC cell lines were treated with low concentration of docetaxel (0.1 nM) and doxorubicin (10 nM) for 24h, and western blot results showed that the 24h treatment had little effect on P4HA1 protein levels.

Reviewer comments: Figure S5 is not properly referenced. The authors should recheck the consistency of the figure calls and numbers both in text and in figure legends since the reviewers have found several inconsistencies.

Response: We apologize for these oversights. The reference of figures has been rechecked and corrected.

Reviewer #2, Expertise: Prolyl hydroxylase, metabolism (Remarks to the Author):

The authors correctly addressed my comments. I have no further questions.

Reviewer #3, Expertise: Breast cancer, drug resistance (Remarks to the Author):

Authors have addressed most concerns satisfactorily. Two minor concerns remains:

Reviewer comments: 1) In response to the concern that P4HA1 staining in metastatic vs primary tumors should be conducted to test whether P4HA1 levels associate with TNBC progression, authors argue that since P4HA1 is associated with EMT, they do not expect to identify higher P4HA1 levels in metastatic sites. While this may be true, such a claim would be best substantiated by analysis demonstrating that P4HA1 levels in primary tumors associate with a higher likelihood of metastatic progression. Including this analysis will greatly strengthen the clinical relevance of this manuscript.

Response: This point is well taken. We have performed the Kaplan Meier analysis to determine the association between cancer metastasis and P4HA1 expression in primary tumors using the published microarray dataset (49). We found that increased P4HA1 mRNA levels were associated with a short distant metastasis free survival in breast cancer patients and basal-like breast cancer patients. We have included these results on page 11 and in supplemental Figure (Figure S6a).

Reviewer comments: 2) In response to the request that the authors show stem cell markers enrichment (CD44/CD24low) in in vivo tumors pre and post treatment with docetaxel, authors present ALDH data. However, the use of ALDH as a stem cell marker has been controversial and CD24/44 levels are considered the gold standard in the field. We recommend the authors assess CD24/44 rather than ALDH levels in this figure.

Response: Following the reviewer's advice, we now include the CD24 and CD44 IHC staining in supplemental figure 7 (Figure S7a).

Reviewers' Comments:

Reviewer #1:

Remarks to the Author:

The authors have now addressed all my remaining comments.

Reviewer #3:

Remarks to the Author:

authors have satisfactorily responded to issues raised